# Coordination of two opposite flagella allows high-speed swimming and active turning of individual zoospores

Quang D Tran[1]*[†], Eric Galiana[2], Philippe Thomen[1], Céline Cohen[1], François Orange[3], Fernando Peruani[4,5], Xavier Noblin[1]*

[1]Université Côte d'Azur, CNRS UMR 7010, Institut de Physique de Nice (INPHYNI), Nice, France; [2]Université Côte d'Azur, INRAE UMR 1355, CNRS UMR 7254, Institut Sophia Agrobiotech (ISA), Sophia Antipolis, France; [3]Université Côte d'Azur, Centre Commun de Microscopie Appliquée (CCMA), Nice, France; [4]Université Côte d'Azur, CNRS UMR 7351, Laboratoire J.A. Dieudonné (LJAD), Nice, France; [5]CY Cergy Paris Université, CNRS UMR 8089, Laboratoire de Physique Théorique et Modélisation, Cergy-Pontoise, France

*For correspondence:
duc-quang.tran@ijm.fr (QDT);
xavier.noblin@univ-cotedazur.
fr (XN)

Present address: [†]Université
Paris Cité, CNRS, Institut Jacques
Monod, Paris, France

Competing interest: The authors
declare that no competing
interests exist.

Reviewing Editor: Raymond
E Goldstein, University of
Cambridge, United Kingdom

**Abstract** *Phytophthora* species cause diseases in a large variety of plants and represent a serious agricultural threat, leading, every year, to multibillion dollar losses. Infection occurs when their biflagellated zoospores move across the soil at their characteristic high speed and reach the roots of a host plant. Despite the relevance of zoospore spreading in the epidemics of plant diseases, individual swimming of zoospores have not been fully investigated. It remains unknown about the characteristics of two opposite beating flagella during translation and turning, and the roles of each flagellum on zoospore swimming. Here, combining experiments and modeling, we show how these two flagella contribute to generate thrust when beating together, and identify the mastigonemes-attached anterior flagellum as the main source of thrust. Furthermore, we find that turning involves a complex active process, in which the posterior flagellum temporarily stops, while the anterior flagellum keeps on beating and changes its gait from sinusoidal waves to power and recovery strokes, similar to *Chlamydomonas*'s breaststroke, to reorient its body to a new direction. Our study is a fundamental step toward a better understanding of the spreading of plant pathogens' motile forms, and shows that the motility pattern of these biflagellated zoospores represents a distinct eukaryotic version of the celebrated 'run-and-tumble' motility class exhibited by peritrichous bacteria.

## Editor's evaluation

The authors present a study of the swimming behaviour of the zoospores of the water mold *Phytophthora* (Greek "Plant Destroyer"), which is responsible for significant crop damage worldwide. The motility of the zoospores is likely a significant contributor to the successful spread of the disease, and as such its study has potential wide impact. The authors suggest using a model that the anterior "hairy" (covered with mastigonemes) flagellum is the primary contributor to motility, and show with high-speed imaging that the microorganism is able to turn on the spot by stopping its posterior flagellum, and changing the beat-pattern of its anterior flagellum from "sperm-like" to "*Chlamydomonas*-like".

## Introduction

Life of swimming microorganisms in viscosity-dominant world has been of great interest in biophysics research. The problems on microbial locomotion of those tiny individual flagellated swimmers are still far to be fully understood. There have been a multitude of theoretical and experimental models of

**eLife digest** Microorganisms of the *Phytophthora* genus are serious agricultural pests. They cause diseases in many crops, including potato, onion, tomato, tobacco, cotton, peppers, and citrus. These diseases cause billions of dollars in losses each year. Learning more about how the tiny creatures disseminate and reach host plants could help scientists develop new ways to prevent such crop damage.

The spore cells of *Phytophthora*, also known as zoospores, have two appendages called flagella on their bodies. A tinsel-shaped flagellum is near the front of the creature and a long smooth filament-like flagellum is near the posterior. Zoospores use their flagella to swim at high speeds through liquid toward potential plant hosts. Their complex swimming patterns change in response to different physical, chemical, and electrical signals in the environment. But exactly how they use their flagella to generate these movements is not clear.

Tran et al. reveal new details about zoospore locomotion. In the experiments, Tran et al. recorded the movements of zoospores in a tiny 'swimming pool' of fluid on top of a glass slide and analyzed the movements using statistical and mathematical models. The results uncovered coordinated actions of the flagella when zoospores swim in a straight line and when they turn. The tinsel-like front flagellum provides most of the force that propels the zoospore forward. To do this, it beats with an undulating wave pattern. It shifts the beating to a breast-stroke pattern to change direction. The posterior flagellum provides a smaller forward thrust and temporarily pauses during turns.

The study provides new details about zoospore's movements that may help scientists develop new strategies to control these pests. It also offers more information about how flagella coordinate their actions to switch speeds or change directions that may be of interest to other scientists studying organisms that use flagella to move.

microswimmers that study the hydrodynamics of the individual and collective motions of those cells (*Ghanbari, 2020*; *Lauga and Goldstein, 2012*). These swimming cells can be categorized into two groups: eukaryotes (having nuclei) and prokaryotes (no nuclei). *Escherichia coli* is one of the most studied prokaryotic swimmers, which possesses a bundle of passive helical flagella controlled by a rotary motor attached to the cell body (*Turner et al., 2000*). Eukaryotic microswimmers, such as green algae *Chlamydomonas* (*Mitchell, 2001*; *Polin et al., 2009*) and spermatozoa (*Ishijima et al., 1986*; *Mortimer et al., 1997*), have active and flexible flagella along which molecular motors are distributed. Here, we introduce a new type of microswimmer, named *Phytophthora* zoospores, which has two different flagella collaborating for unique swimming and turning mechanisms (*Figure 1(A)*).

*Phytophthora* is a genus of eukaryotic and filamentous microorganisms. They are classified as oomycetes and grouped in the kingdom of the Stramenopiles with the heterokont algae (such as diatoms and brown algae) (*Burki, 2014*; *Derelle et al., 2016*). A number of *Phytophthora* species are plant pathogens and cause tremendous damages to agro- and eco-systems (*Derevnina et al., 2016*; *Kamoun et al., 2015*). Nowadays, *Phytophthora* diseases are responsible for a big impact on economies with billions of dollars of damages each year and remain a threat to the food security worldwide (*Drenth and Sendall, 2004*; *Judelson and Blanco, 2005*; *Strange and Scott, 2005*). The diseases are pervasive as they release swimming biflagellated spores called 'zoospores' which initiates the spreading through water. These zoospores are able to achieve speed up to $250\,\mu\mathrm{m\,s^{-1}}$(*Appiah et al., 2005*) through thin water films, water droplets on leaves, or through pores within moist soils. To facilitate the spreading, their cell bodies store an amount of energy (mycol-aminarin, lipid) allowing them to swim continuously for several hours (*Judelson and Blanco, 2005*). In natural ecosystems and even more in agro-systems, putative host plants are usually close. This proximity makes the distance to find a plant relatively short and it is compatible to the time-ability of zoospores to swim. When the zoospores reach plant roots, they stop swimming and release their flagella to produce a primary cell wall and become germinative cysts which are able to penetrate into the host tissue. Then, they start a hyphal growth inside the infected plant. In this study, we investigate the telluric species *P. parasitica*, a polyphagous pathogen attacking a wide range of hosts such as potato, onion, tomato, tobacco, ornamentals, cotton, pepper, citrus plants, and forest ecosystems (*Panabieres et al., 2016*).

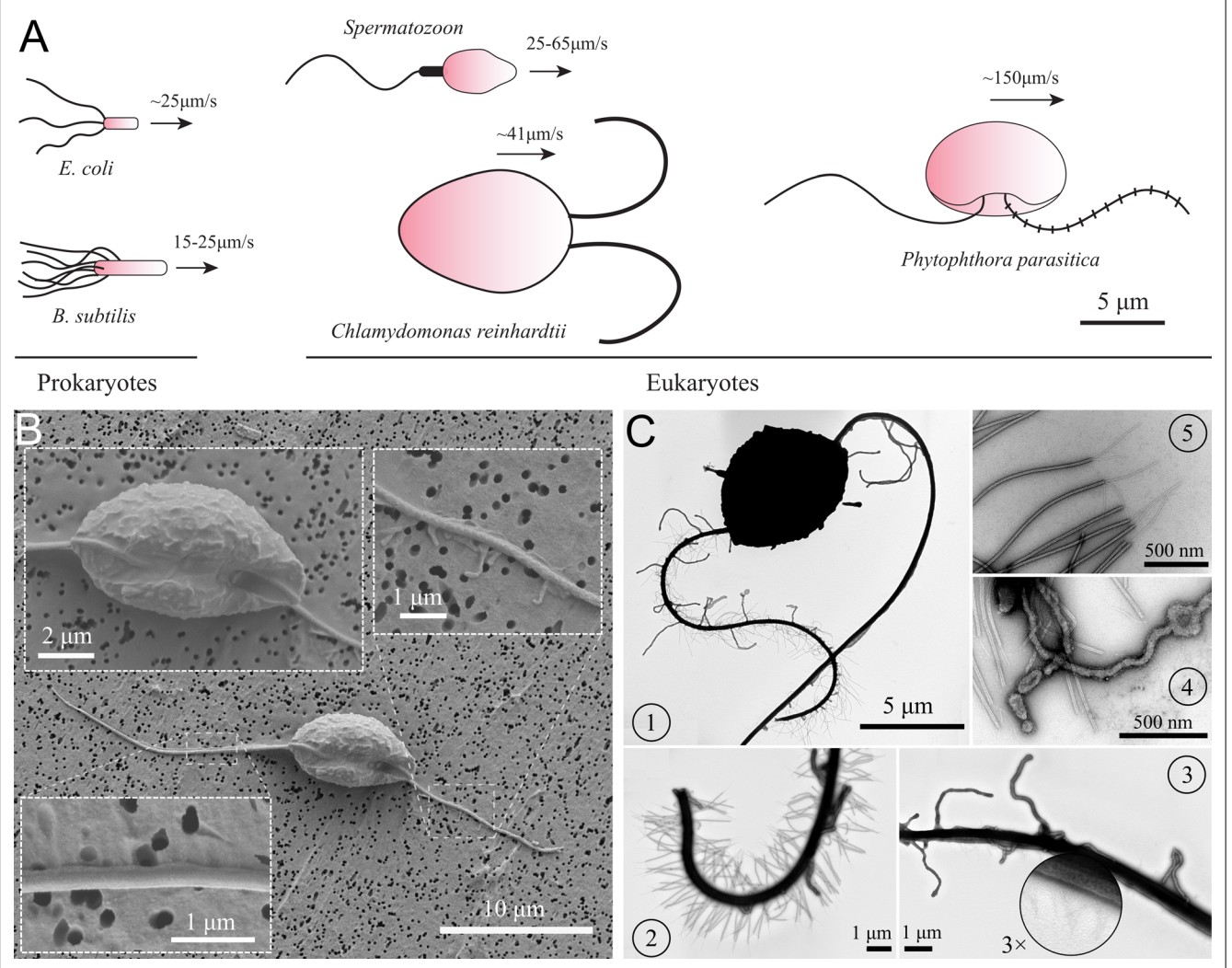

**Figure 1.** Characteristics of *P. parasitica* zoospore. (**A**) Swimming of zoospores in comparison with different prokaryotic and eukaryotic microswimmers. Black arrows indicate the swimming direction of the swimmers. (**B**) Scanning electron microscopy images of the zoospore. The insets show the enlarged images of the cell body and the two flagella. (**C**) Transmission electron microscopy images with negative staining. (1) Image of the zoospore showing the different structures of the two flagella. The anterior has multiple mastigonemes, while the posterior has a smooth straight structure. (2) Close zoom-in image of the anterior flagellum. It is noticed that there are two types of mastigonemes on this flagellum: one with straight tubular shape, the other with non-tubular shape but longer and bigger in size. (3) Close zoom-in image of the posterior flagellum. There are plenty of thin and short hairs wrapping along the flagellum and several non-tubular mastigonemes appearing near the cell body. (4) The non-tubular mastigonemes. (5) The tubular mastigonemes with tiny hairs at the tips.

Previous studies have shown that during the spreading and approaching the host, zoospores can have complicated swimming patterns and behaviors as they experience multiple interactions with environmental signals, both physical, electrical and chemical, in soil and host-root surface (*Bassani et al., 2020a*). Near the plant-root, zoospores can perceive various stimuli from the environment, such as ion exchange between soil particles and plant roots, the chemical gradients generated by root exudates, which activate cell responses. This results in coordinated behaviors of zoospores, allowing them to preferentially navigate to the water film at the interface between soil particles and plant roots. For instance, potassium, which is uptaken by roots in the soil, reduces zoospore swimming speed, causes immediate directional changes and also results in perpetual circle trajectories (*Appiah et al., 2005*; *Galiana et al., 2019*). Bassani et al. provide transcriptomic studies showing that potassium induces zoospore aggregation, which facilitates the advantages for zoospores to attack the host-root (*Bassani et al., 2020b*). Experimental evidence has demonstrated that zoospore-zoospore interaction can lead to 'pattern swimming', a microbial bioconvection happened without the appearance

of chemical or electrical signals (*Ochiai et al., 2011*; *Savory et al., 2014*). These findings urge for a better understanding of the swimming physics of the individual zoospores and how the combination of their two heterogeneous flagella results in those complex swimming behaviors.

A zoospore is usually about 10 μm in size and has a kidney-like cell body (*Mitchell et al., 2002*; *Walker and van West, 2007*). At least two unique traits distinguish zoospores from other prokaryotic and eukaryotic microswimmers currently studied using physical approaches: (i) The two flagella beat longitudinally along the anterior-posterior axis of the cell body and not laterally as in the case of the green algae *Chlamydomonas*; (ii) two flagella distinguished from each other as the anterior flagellum has a tinsel-like structure, while the posterior flagellum has a smooth whiplash one, both beating periodically with wave propagation directions outwards the body. At first sight, these two flagella seem to be competing each other due to the opposite wave propagation directions. Contrarily, multiple mastigoneme structures on the anterior flagellum of zoospores are shown to have thrust reversal ability, which makes both flagella generate thrust in the same direction and propel the cell body forwards (*Bassani et al., 2020b*; *Cahill et al., 1996*). Although it has been known about how zoospores swim, characteristics of the swimming and the beating flagella have not been statistically reported. The effects of mastigonemes on zoospore swimming also need to be carefully investigated since the mechanical properties of mastigonemes such as size, rigidity, density, can affect the swimming differently (*Namdeo et al., 2011*). For instance, although mastigonemes are shown to generate thrust reversal in *P. palmivora* zoospores (*Cahill et al., 1996*), they do not contribute to enhance swimming of *C. reinhardtii* (*Amador et al., 2020*).

In other microswimmers, their flagella are often synchronized to perform a cooperative swimming when they are a few microns away from each other (*Friedrich and Jülicher, 2012*; *Lauga and Goldstein, 2012*). For examples, *C. reinhardtii* performs breaststroke swimming by two flagella drawing away and back to each other (*Polin et al., 2009*) or *E. coli*, *B. subtilis* bacteria's flagella form a bundle and rotate together like a corkscrew to propel the cell body (*Turner et al., 2000*). The question that whether zoospores, as eukaryotic swimmers, possess the similar cooperative behaviors of their flagella is of good interest. It has been previously claimed that zoospore flagella are independent on each other and able to perform different tasks. *Carlile, 1983* describes that the anterior flagellum is responsible for pulling the zoospore through water whereas the posterior flagellum acts as a rudder for steering the cell. However, *Morris et al., 1995* observe *P. palmivora* zoospores stop momentarily and then self-orientate their bodies to a new direction relatively to the posterior flagellum. Nevertheless, the cooperative actions of the motor and rudder have not been carefully observed nor investigated, which remains unclear about how zoospores change direction either by random walks or in response to chemical and physical environment. This motivates us to unveil the physics behind individual swimming of zoospores.

In this article, we first investigate characteristics of zoospore trajectories at a global scale, then focus on the flagella scale's swimming mechanisms. We observe that zoospores can perform long and stable straight runs, discontinued by active turning events. We obtain statistics of the trajectories and develop a numerical model to study and extrapolate the zoospore spreading characteristics solely by random walks. Then, we detail an in-depth study on the hydrodynamics of *P. parasitica*'s flagella and acquire a mathematical model to correlate the functions of two flagella on the motion of straight runs. Although theoretical models for microswimmers with single mastigonemes-attached flagella have been formulated (*Brennen, 1975*; *Namdeo et al., 2011*), models for microswimmers with two heterokont flagella have yet been considered as in case of zoospores. Here, we use Resistive Force Theory and further develop the model of a single flagellum with mastigonemes (*Brennen, 1975*; *Namdeo et al., 2011*) to adapt it with another smooth flagellum and a cell body, using a hypothesis of no interactions between two flagella. Moreover, we discover a unique active turning mechanism of zoospores including a body rotation then steering to a new direction, which results from the instantanous gait changing ability of their anterior flagellum. Our study reveals the mechanism and characteristics of zoospore spreading, which provides better insights on understanding and control of *Phytophthora* diseases.

## Results and discussion

### Characteristics of *P. parasitica*'s cell body and flagella

To understand the swimming, we first look at the cell body and flagellar structures of *P. parasitica*. By using Scanning Electron Microscopy (EM), we are able to observe the shape of the cell body and the positions of the flagellar base (*Figure 1(B)*). The cell in general has an ellipsoidal shape with tapered heads and a groove along the body. The size of the body is measured to be $8.8 \pm 0.4\,\mu m$ (SEM) in length, and $4.7 \pm 0.1\,\mu m$ (SEM) in width. The anterior flagellum attaches to the cell body in a narrow hole at one side of the groove, possessing an average length of $15.5 \pm 0.1\,\mu m$ (SEM). The posterior flagellum has the same diameter as the anterior's ($0.3\,\mu m$) but it is longer ($20.3 \pm 0.76\,\mu m$ (SEM)), and attaches directly to the surface of the groove. The roots of two flagella are apart from each other with a distance of $2.9 \pm 0.1\,\mu m$ (SEM). Some mastigoneme structures were observed on the anterior flagellum, but could only be distinguished with difficulty by Scanning EM technique.

Observing the zoospores with Transmission Electron Microscopy (TEM) after negative staining, we discover multiple mastigonemes on both flagella (*Figure 1* (C1-3)). There are two different types of mastigonemes on the anterior flagellum: (type-1) straight and tubular shape, high density (~ 13 per $\mu m$), $0.03\,\mu m$ diameter, $1.5\,\mu m$ long; (type-2) curved and irregular shape, longer and thicker in size ($0.1\,\mu m$ diameter, $1.8\,\mu m$ long), and randomly distributed (*Figure 1* (C2)). The posterior flagellum instead has a smooth whip shape with plenty of very fine hairs on the surface (*Figure 1* (C3)). These hairs wrap around the flagellum to increase the contact surface, thus increasing the propulsion efficiency (*Lee, 2018*). We also see a few type-2 mastigonemes on the posterior flagellum but they only appear near the root. The function of type-2 mastigonemes (*Figure 1* (C4)) is unknown, but their flexibility and random arrangement suggest that they might not contribute to generate drag. In contrary, the type-1 mastigonemes (*Figure 1* (C5)) are tripartite hairs that occur in most of Stramenopiles kingdom. They are known to be able to generate increased drag and reverse thrust for the anterior flagellum (*Cahill et al., 1996*; *Dodge, 2012*).

### Statistics of individual swimming patterns

We investigate the characteristics of zoospore swimming by analyzing their trajectories and behaviors in water. To facilitate that, we perform microscopic assays where a low concentration of individual zoospores are released to an open thin film of water with thickness of ~100 μm on a glass slide. The setup of the water thin film can be visualized as a 'swimming pool' that is not covered as we want to avoid the unwanted physical interactions of zoospores with the top when they experience aerotaxis. The zoospore swimming is captured at 60 fps (interval time between two consecutive frames, $\Delta t \approx 0.0167\,s$). The images are processed by Fiji (*Schindelin et al., 2012*) and Trackmate plugin (*Tinevez et al., 2017*) to semi-automatically track the positions of zoospores during the experiment duration (see *Video 1*). *Figure 2(A)* illustrates the trajectories of zoospores captured from the microscopic assay. These trajectories indicate that zoospore can perform long and straight runs, some can even cross the whole observed region. The straight runs are separated by multiple turning events when zoospores randomly change directions. With this swimming strategy, zoospores can be categorized as run-and-tumble active particles (*Khatami et al., 2016*). From the position data of each zoospore over time, we achieve its movement characteristics defined by two parameters: magnitude of speed $U$ and moving directions $\theta$ (*Figure 2(B–C)*), after applying moving average method with step length $n = 12$ to improve the accuracy of moving direction and instantaneous speed of the zoospore. From $U$ values, we can separate the movement of zoospores into 2 states: running state during straight runs and stopping state at turning events. While running,

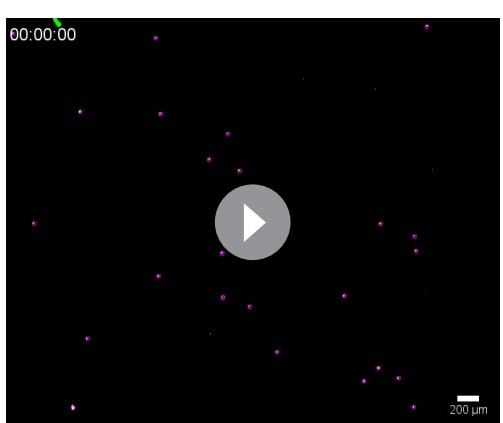

**Video 1.** Zoospores swimming in thin water film. The video was captured in the microscopic assay, with objectives 4×, 60 fps, duration 60 s. The trajectories of zoospores were tracked by combining TrackMate (in Fiji) and manual tracking.

https://elifesciences.org/articles/71227/figures#video1

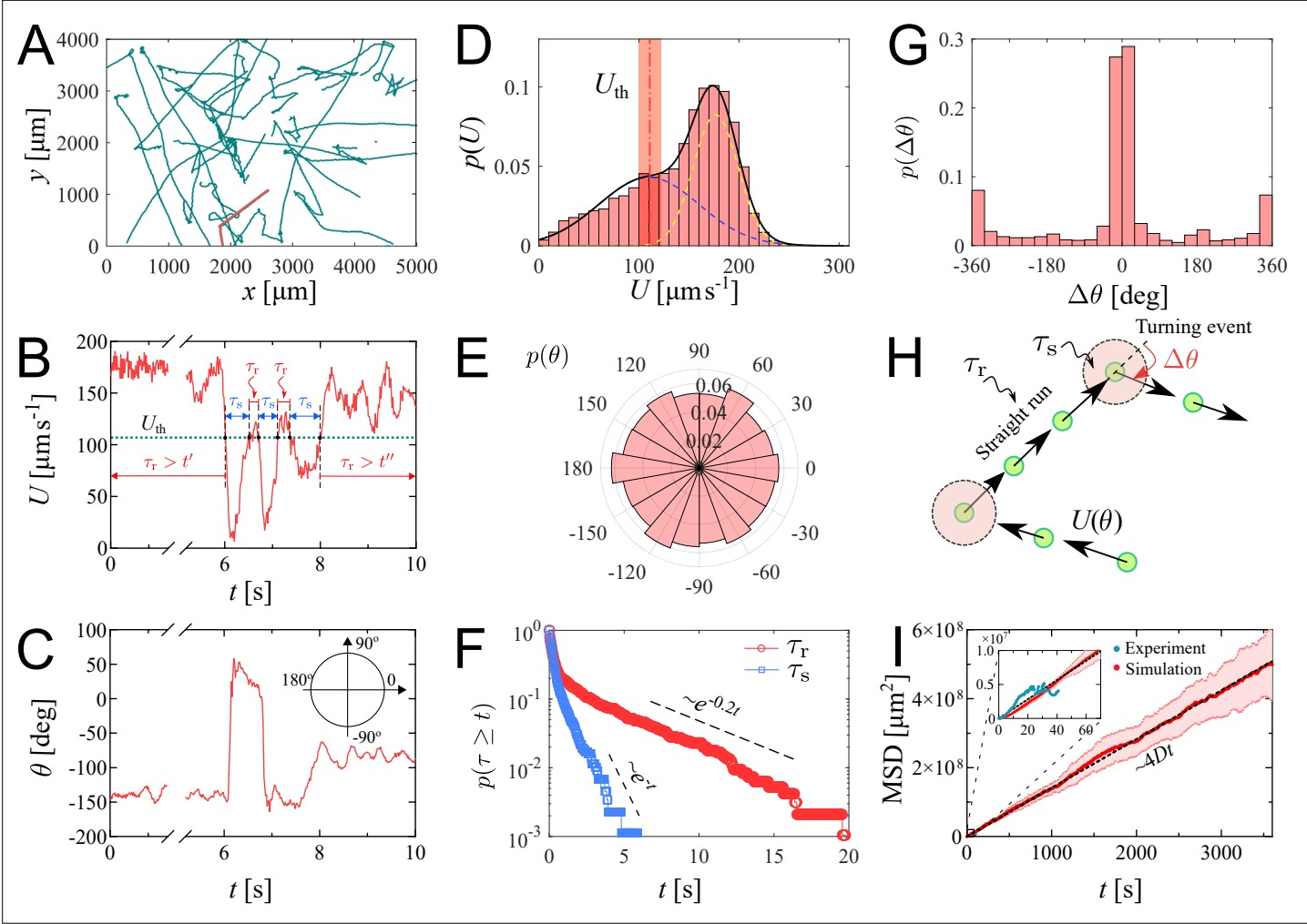

**Figure 2.** Swimming trajectories of *P. parasitica* zoospores. (**A**) Trajectories of zoospores swimming in water captured from the microscopic assay for 60 s. Sample size $N = 58$. Note: not all trajectories are shown. Each position of the zoospores is captured every $\Delta t = 0.0167$ s. The trajectories are smoothed with moving average (step length $n = 12$). (**B**) The progression of speed $U$ and (**C**) moving directions $\theta$ over time of a single zoospore extracted from the population in the assay. (**D**) Distribution of zoospore speed $p(U)$. (**E**) Polarity distribution of moving direction $p(\theta)$. (**F**) Survival curves $p(\tau \geq t)$ of the running time $\tau_r$ and stopping time $\tau_s$. (**G**) Distribution of turning angle $p(\Delta\theta)$, with positive angle values indicating counter-clockwise, and negative as clockwise direction. (**H**) Schematics showing the strategy of the simulation model of zoospores swimming in water. (**I**) The estimated mean squared displacement (MSD) over time intervals $t$, constructed from the simulation data. The inset compares the experimental data and simulation of MSD at the experimental time-scale of 60 s. By simulation, at long time scale of 1 h, MSD of zoospores shows a diffusion of Brownian particles with the diffusion coefficient $D = 3.5 \times 10^{-4} \text{cm}^2 \text{s}^{-1}$.

$U$ and $\theta$ vary around a constant value. At turning events, $U$ drops drastically, (occasionally close to 0) then quickly recovers, $\theta$ also rapidly changes to a new value. $U_{th}$ is defined as the threshold speed that separates the two states of running and turning. With $U_{th}$, we determine two important parameters of zoospore swimming: (i) running time $\tau_r$ as the duration when $U \geq U_{th}$, deciding how long a zoospore is able to travel without turning; (ii) stopping time $\tau_s$ as the duration, when $U \leq U_{th}$, for a zoospore to perform a turn.d

We plot the distribution of $U$ for all the trajectories of zoospores swimming in the observed region for duration of 60 s (total number of zoospores $N = 58$) in *Figure 2(D)*. The speed distribution $p(U)$ exhibits a combination of two different normal distributions: $f_1(U) = 0.082\,e^{-(U-\mu_1)^2/2\sigma_1^2}$ with $\mu_1 = 176.6$, $\sigma_1 = 22.3\,\mu\text{m s}^{-1}$, and $f_2(U) = 0.043\,e^{-(U-\mu_2)^2/2\sigma_2^2}$ with $\mu_2 = 110$, $\sigma_2 = 53\,\mu\text{m s}^{-1}$. This bimodal distribution of $U$ indicates that zoospore speed fluctuates around two speed values $U = \mu_1$ and $U = \mu_2$, corresponding to two behavioral states of running and turning. The distribution $f_1$ is associated with the running state where zoospores experience stable moving speed, while $f_2$ represents

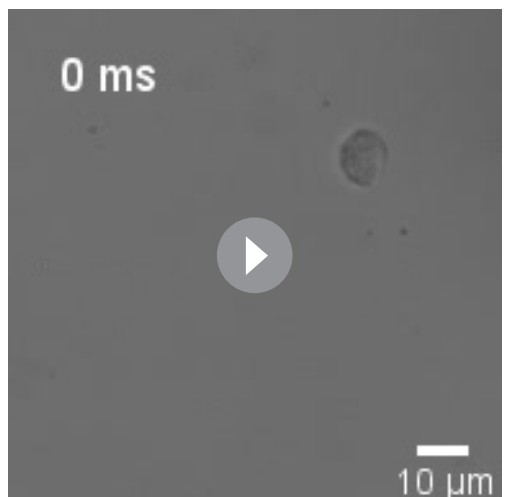

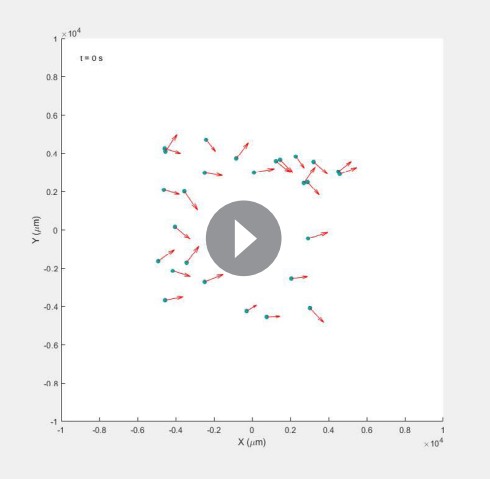

**Video 2.** A zoospore swimming near water/air interface.
https://elifesciences.org/articles/71227/figures#video2

**Video 3.** The result from our simulation with the strategy described in Figure 2(H). The parameters of the simulation are extracted from the statistics of the swimming of zoospores, presented in Figure 2(D–G).
https://elifesciences.org/articles/71227/figures#video3

the speed at turning events where zoospores reduce their speed from a stable running speed to 0 then quickly recover. We achieve the fitting curve for $p(U)$, resulting from the sum of two Gaussian fits $f_1 + f_2$, and choose the speed at inflection point of the fitting curve where $\mu_s \leq U \leq \mu_r$ as the speed threshold to separate two behavioral states, $U_{th} = 111.5\,\mu m\,s^{-1}$ at the inflection point determines a turning point for a significant change of the speed from running to turning state. The sensitivity of our $U_{th}$ selection can be tolerated by ±10 % of the chosen value, ranging from 100 to 122.5 $\mu m\,s^{-1}$ (See Appendix 5). We also plot the polarity distribution of zoospores in *Figure 2(E)* based on the moving direction $\theta$ and acquire an equally distributed in all directions. With the defined $U_{th}$, we calculate and plot the distributions of $\tau_r$ and $\tau_s$ in form of survival curves $p(\tau \geq t)$ in *Figure 2(F)*. Both survival curves show complex behaviors of zoospores during running and turning. The statistics $p(\tau_s \geq t)$ can be considered as sum of two exponential decays with average stopping time $\bar{\tau}_s = 0.37\,s$. Also, $p(\tau_r \geq t)$ is in form of two exponential decays with average running time $\bar{\tau}_r = 1.0\,s$. We can estimate that zoospores stop and turn with the frequency $1/\bar{\tau}_r$ greater than 1 Hz ($p(\tau_r < 1.0\,s) = 0.82$). Based on the moving direction over time, we calculate the average turning speed of zoospores at $\bar{\dot{\theta}} \approx 0.6\pi\,rad\,s^{-1}$. At each stopping time they perform a turning angle $\Delta\theta = \theta_i - \theta_e$, where $\theta_i$ and $\theta_e$ is the moving direction right before and after each turning event, respectively. The distribution of $\Delta\theta$ is shown in *Figure 2(G)*, demonstrating the equal preference of turning directions. It is also shown that zoospores preferentially turn with the angle around 0°, which we speculate that it results from the failed out-of-plane movement when zoospores swim near the water/air interface during their aerotaxis (See *Video 2*).

Since the motion of zoospore is characterized by the succession of straight runs and turning events, as illustrated in *Figure 2(H)*, in order to quantify their large-scale transport properties, we assemble all previous measurements in the following way. Each straight run is characterized by a speed $U_r > U_{th}$ and a duration $\tau_r$ drawn from the distributions in *Figure 2(D), and (F)*, respectively. After a run phase, an idle phase of duration $\tau_s$, drawn from *Figure 2(F)* follows. The moving direction of the $r$-th run phase is given by $\cos(\theta_r)\hat{x} + \sin(\theta_r)\hat{y}$, i.e. it is parameterized by an angle $\theta_r$. Note that moving direction of two consecutive $r$-th and $(r + 1)$-th run phases are correlated. Moreover, $\theta_{r+1} = \theta_r + \Delta\theta$, where $\Delta\theta$ is a random angle drawn from the distribution in *Figure 2(G)*. Mathematically, the position $\mathbf{x}_m$ of the zoospore after $m$ run phases is given by $\mathbf{x}_m = \sum_{r=1}^{m} U_r \tau_r[\cos(\theta_r)\hat{x} + \sin(\theta_r)\hat{y}]$, and its mean-square displacement is $\text{MSD}(m) = \langle(\mathbf{x}_m - \langle\mathbf{x}_m\rangle)^2\rangle$, where $\langle \cdots \rangle$ denotes average over realizations of the process. Our simulation results in a diffusive behavior of zoospores, with the MSD proportional to $t$ (see *Video 3*). The diffusion coefficient is then obtained from $D = \lim_{m\to\infty} \frac{\text{MSD}(m)}{4m(\langle\tau_r\rangle + \langle\tau_s\rangle)}$. In the computation of MSD and $D$, we assume that the only random variable exhibiting correlations is $\theta_r$, while $U_r$, $\tau_r$, and $\tau_s$ are uncorrelated. This procedure allows us to obtain an estimate of $D$, with

$D = 3.5 \times 10^{-4} \mathrm{cm}^2\,\mathrm{s}^{-1}$, which is in the same order of magnitude as the diffusion coefficient of *C. reinhardtii*'s (*Polin et al., 2009*). We stress that direct measurements of $D$ based on experimental MSD data are highly unreliable given the relatively small number of trajectories and their short duration. In our case, the data are enough to obtain reliable estimates on the distributions of $\Delta\theta$, $U_r$, $\tau_r$, and $\tau_s$ from which, as explained above, $D$ can be reliably estimated from simulations. Similar methods of using a theoretical random walk model to estimate the macroscopic parameters from the microscopic experiment have been previously developed for *C. reinhardtii* (*Hill and Hader, 1997*; *Vladimirov et al., 2004*). We emphasize that $D$ represents the estimation of diffusion coefficient of individual swimming of zoospores from random walks. This is more to show the intrinsic ability of individual zoospores to perform spatial exploration, rather than to quantify the bulk diffusivity where the collective swimming behaviors, which involve zoospore-zoospore interactions, play a major role.

## Role of two flagella in swimming motions of zoospores

Our statistics study on swimming trajectories of zoospores has delivered characteristics of their movement at large-scale, including the straight runs and turning events. These motions are controlled by two flagella oriented in opposite directions along the cell body's anterior-posterior axis. In this section,

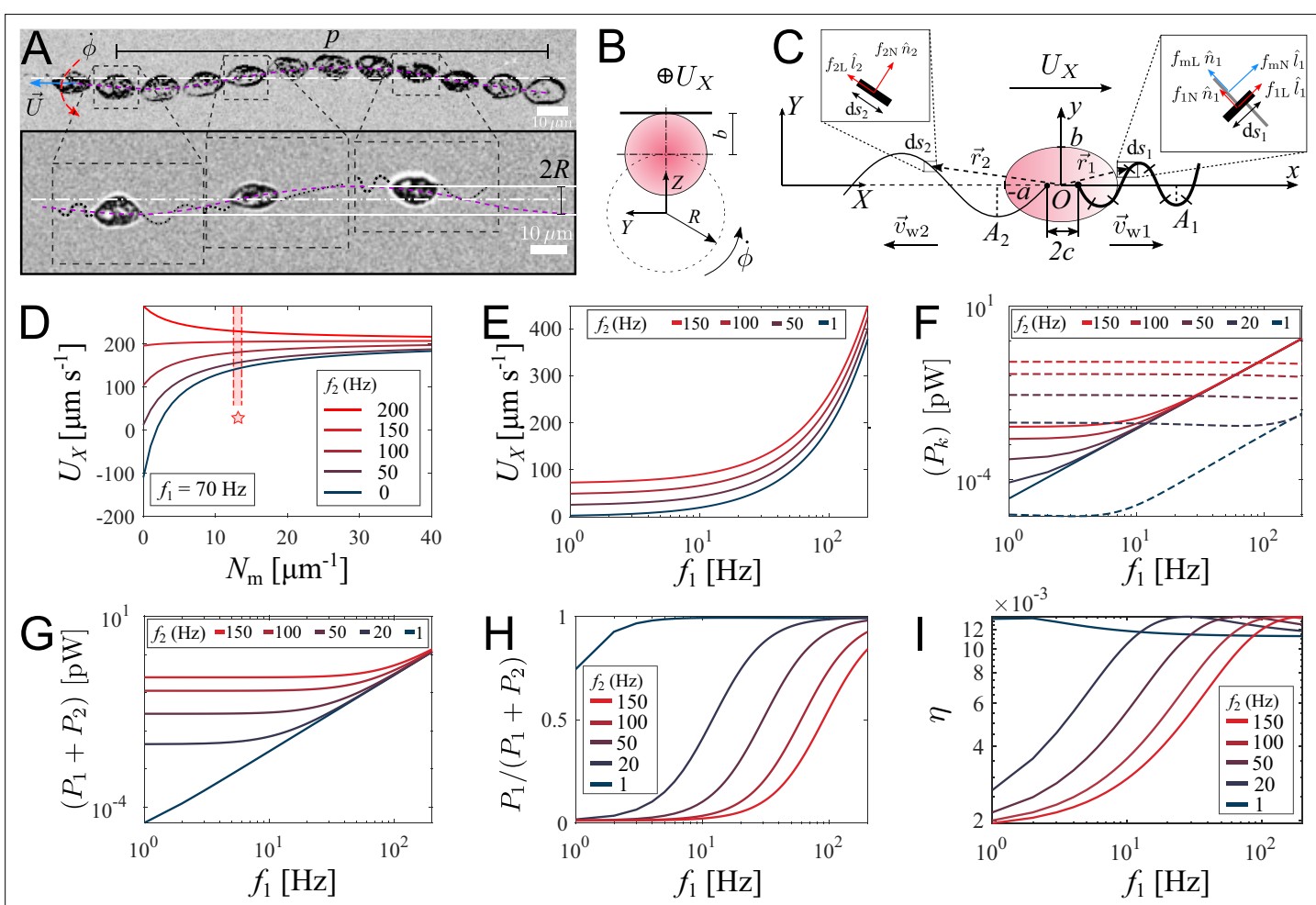

**Figure 3.** Theoretical model of swimming individual *P. parasitica* zoospore. (**A**) Images of an individual zoospore swimming with two flagella beating in sinusoidal waveform shapes and its cell body gyrating with rate $\dot{\phi}$ while moving forward with speed $U$. The combined motion results in a helical swimming trajectory with pitch $p$ and radius $R$. (**B**) Schematics showing the gyration of the cell body. (**C**) Theoretical model of a zoospore translating in a 2D plane using Resistive Force Theory. (**D**) The dependence of translational speed $U_X$ on the type-1 mastigoneme density ($N_\mathrm{m}$). The range of $N_\mathrm{m}$ with symbol (⋆) indicates the values measured by TEM. (**E**) The effects of beating frequencies of the two flagella, $f_1$ and $f_2$, on zoospore speed $U_X$, (**F**) power consumption of each flagellum $P_k$, (**G**) total power consumption of two flagella ($P_1 + P_2$), (**H**) power distributed to the anterior flagellum and (**I**) propelling efficiency of both flagella $\eta$. In these plots, $N_\mathrm{m}$ is set at 13 $\mu\mathrm{m}^{-1}$.

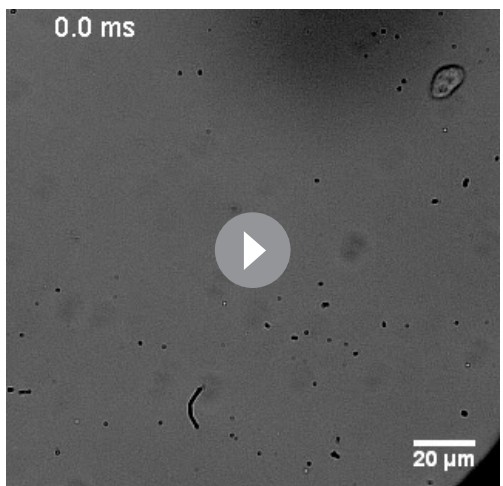

**Video 4.** An individual zoospore swimming in water. First, it swims straight with the helical trajectory due to the spontaneous gyration of the cell body, then performs a 180° turning.

https://elifesciences.org/articles/71227/figures#video4

we look in-depth to how these two flagella together generate speed and perform turning for zoospores by conducting microscopic assays at small length scale, of which the flagella are visible and in very short time scale.

## Straight runs

We record movement of zoospores during their straight runs with visible flagella by conducting brightfield microscopy with 40× objective and a high-speed camera capturing at 2000 fps at exposure time 200 $\mu$s. In *Figure 3(A)*, we show images of a *P. parasitica* zoospore swimming by two flagella beating in sinusoidal shapes with the wave propagation in opposite directions. While translating, the cell body gyrates around the moving direction simultaneously, which results in a helical swimming trajectory (see *Video 4* for a long run of a zoospore swimming in water). We believe that this gyrational motion might result from the intrinsic chiral shape of the zoospore body and off-axis arrangement of their flagella (*Figure 1(B)*). Indeed, previous studies have shown that chirality of a microswimmer's body induces spontaneous axial rotation resulting from the translational motion (*Keaveny and Shelley, 2009*; *Löwen, 2016*; *Namdeo et al., 2014*). From multiple observations, we obtain the pitch and radius of the helical trajectories at $p = 130 \pm 8\,\mu$m (SEM) and $R = 4.0 \pm 0.2\,\mu$m (SEM), respectively (data presented in Appendix). We then estimate the gyrational speed of the cell body $\dot{\phi} = 2\pi/\Delta t_p$, where $\Delta t_p$ is the duration the zoospore travels through a full turn of the helical path (*Figure 3(B)*). We obtain $\dot{\phi} = (3.6 \pm 0.3)\pi\,\mathrm{rad\,s^{-1}}$ (SEM) (see Appendix). The observations of helical trajectories also confirm that each flagellum of zoospores beat as a flexible oar in a 2D plane as we notice the two flagella flattened into two straight lines during the gyration. Thus, zoospores are not expected to swim in circles when interacting with no-slip boundaries as seen in case of *E. coli* with a rotating flagellar bundle. The 'curved straight runs' of zoospores that we observed in *Figure 2(A)* might result from rotational diffusion and thermal fluctuations.

We retrieve the parameters of the beating flagella including beating frequencies $f$ and wavelengths $\lambda$, by applying kymographs on the cross-sections within the two flagella and normal to the moving direction of the zoospore. We present more details of the parameter retrieval with kymographs in Appendix. Our data show that when swimming in straight runs, the anterior flagellum of zoospores usually beats at $f_1 \approx 70\,\mathrm{Hz}$ while the posterior flagellum beats with the frequency $f_2 \approx 120\,\mathrm{Hz}$ that is approximately 1.7-fold faster than the anterior's. We focus on a very short duration of less than 50 ms, which is equivalent to a translation of less than 10 μm. Compared with the rotation motion at time scale of 1 s, we neglect the effect of the cell body's rotation in this short duration.

We then develop a mathematical model to study how the dynamics of beating flagella helps generating thrust for the cell to move forward. We assume that the gyration of the cell body does not affect the shapes and motions of the flagella since the beating frequencies of the two flagella are much higher than the gyrational speed. The gyration also does not contribute to the translation as we consider it as a passive motion resulting from the chirality of zoospores. Thus, the swimming zoospore can be considered as a 2D model (*Figure 3(C)*) in which the cell body is an ellipse defined as $x^2/a^2 + y^2/b^2 = 1$ in its body-fixed frame ($xOy$), with the anterior and posterior flagellum having sine waveform shapes defined as

$$y(x,t) = A_k \sin\left(\omega_k t + (-1)^k \frac{2\pi\left[x + (-1)^k c\right]}{\lambda_k}\right) \tag{1}$$

as $(-1)^k x \leq -c$, where $A_k$ is the amplitude, $\omega_k$ is the angular speed, $\lambda_k$ is the wavelength of the anterior ($k = 1$) and the posterior flagellum ($k = 2$). The two flagella are attached to the cell body at two points both lying on $x$-axis and distanced to the origin $O$ a gap of $c$, and beating with the wave propagation $\vec{v}_w$ having directions against each other. The anterior has length $L_1$ and diameter $d_1$ while those of the posterior are $L_2$ and $d_2$. Additionally, there are multiple tubular type-1 mastigonemes with length $h$ and diameter $d_m$ attached to the surface of the anterior flagellum with density $N_m$ indicating the number of mastigonemes attached on a unit length of the flagellum. It is important to determine the flexibility of these mastigonemes as it would impact the ability of the mastigonemes to produce drag. We estimate the flexibility by a dimensionless parameter, which has been carefully characterized in previous studies (*Khaderi et al., 2009*; *Namdeo et al., 2011*), $F_m = 12\mu K A \omega h^3/(E d_m^3)$, where $\mu$ is the fluid viscosity, $K = 2\pi/\lambda$ is the wave number, $E$ is the Young modulus of the mastigonemes. With this estimation, if $F_m < 0.1$, mastigonemes are considered as fully rigid. In case of zoospores' mastigonemes, we achieve $F_m$ at order of $10^{-4}$, which is much lower than 0.1. Thus, we can assume that the mastigonemes of zoospores are non-deformable and rigidly attached to the anterior flagellum. As a result, hydrodynamic interactions between neighboring mastigonemes can also be neglected. Additionally, we ignore the effects of the type-2 mastigonemes in producing drag due to their non-tubular and random structures.

Zoospores swim in water with very low Reynolds number ($Re << 1$), resulting in negligible inertia, dominant viscous force and the kinetic reversibility (*Purcell, 1977*; *Qiu et al., 2014*). Microswimmers with flexible flagella generate thrust from drag force acted by fluid on the flagellum segments. In our model, we use Resistive Force Theory (RFT) to deal with the calculation of fluid's drag force on the two flagella of the zoospore. RFT has proven to be an effective and accurate method to predict the propulsive force and velocity of microswimmers regardless of the interactions of flagellum-flagellum or flagellum-body (*Gray and Hancock, 1955*; *Lauga and Powers, 2009*; *Marcos et al., 2014*). In case of zoospores where two flagella are in opposite directions, and the flagellum-body interaction is insignificant, RFT is a suitable solution to apply. Following this method, each flagellum is divided into an infinite numbers of very small segments with length $ds_k$, and each segment is located in the body-fixed frame ($xOy$) by a position vector $\vec{r}_k = x_k \vec{i} + y_k \vec{j}$, where $\vec{i}$, $\vec{j}$ are the unit vectors in $x$- and $y$-direction, respectively; $x_k$ and $y_k$ satisfy the shape equation (*Equation 1*).

RFT states that the drag force by fluid acting on an infinitesimal segment $ds$ of the flagellum is proportional to the relative velocity of fluid to the flagellum segment (*Gray and Hancock, 1955*; *Hancock, 1953*; *Koh and Shen, 2016*), as follows

$$\frac{d\vec{F}}{ds} = K_N V_N \hat{n} + K_L V_L \hat{l},$$ (2)

where $V_N$ and $V_L$ are two components of relative velocity of fluid in normal and tangent direction to the flagellum segment, $K_N$ and $K_L$ are drag coefficients of the flagellum in normal and tangent to the flagellum segment, $\hat{n}$ and $\hat{l}$ are the unit vectors normal and tangent to the flagellum segment, respectively. The drag coefficients $K_N$ and $K_L$ are estimated by *Brennen and Winet, 1977*, which depends on fluid viscosity, the wavelength and diameter of a flagellum.

We then apply RFT on each flagellum of the zoospore to calculate the total drag force acting on it. For the posterior flagellum, each segment $ds_2$ is a simple smooth and slender filament (see inset $ds_2$ in *Figure 3(C)*), having drag coefficients $K_{N2}$ and $K_{L2}$. For the anterior flagellum, each segment $ds_1$ contains additional $N_m ds_1$ mastigonemes. Using a strategy from previous models for flagella with mastigonemes (*Brennen, 1975*; *Namdeo et al., 2011*), we consider these mastigonemes stay perpendicular to the segment itself (see inset $ds_1$ of *Figure 3(C)*) and also act as slender filaments experienced drag from water. Interestingly, due to the direction arrangement, the relative velocity normal to the flagellum segment results in drag force in tangent direction to the mastigonemes, and subsequently, the relative velocity tangent to the flagellum segment results in drag force in normal direction to the mastigonemes. In another perspective, we can consider the anterior flagellum receives additional drag from the mastigonemes, which is presented by two increased drag coefficients in normal and tangent direction, defined as

$$K_{N1} = (K_{Nf1} + N_m h K_{Lm1})$$ (3)

and

$$K_{L1} = (K_{Lf1} + N_m h K_{Nm1}), \tag{4}$$

respectively. Here, $K_{Nf1}$ and $K_{Lf1}$ are the drag coefficients in normal and tangent direction of the flagellum filament, respectively; $K_{Nm1}$ and $K_{Lm1}$ are the drag coefficients in normal and tangent direction of the mastigonemes, respectively.

In low Reynolds number condition, total forces equate to zero due to approximately zero inertia. Thus, we derive translational velocity $U_X$ of the zoospore as shown in *Equation 5*

$$U_X = \frac{\frac{2\pi^2 K_{N1} L_1 v_{w1} (\gamma_1 - 1)\beta_1^2}{1 + 2\pi^2 \beta_1^2} - \frac{2\pi^2 K_{N2} L_2 v_{w2} (\gamma_2 - 1)\beta_2^2}{1 + 2\pi^2 \beta_2^2}}{K_{N1} L_1 \left( \frac{\gamma_1 - 1}{1 + 2\pi^2 \beta_1^2} + 1 \right) + K_{N2} L_2 \left( \frac{\gamma_2 - 1}{1 + 2\pi^2 \beta_2^2} + 1 \right) + 6\pi \mu b \xi_e}, \tag{5}$$

where $v_{wk} = \lambda_k f_k$ is the wave propagation velocity of the flagellum, $\gamma_k = K_{Lk}/K_{Nk}$ is the drag coefficient ratio, $\beta_k = A_k/\lambda_k$ is the flagellar shape coefficient, and $\xi_e$ is the shape coefficient of the ellipse cell body (see Appendix for the detailed derivatives).

We first study the effects of mastigonemes on zoospore speed by ploting the value of $U_X$ with different density of the mastigonemes, and varying the beating frequency of the posterior flagellum $f_2$ while the anterior flagellum beats with a usual $f_1 = 70\,\text{Hz}$ in *Figure 3(D)*. The dimensions and physical parameters of the zoospore's cell body and two flagella are shown in Appendix. The plot shows that the appearance of mastigonemes results in a reversed thrust from the anterior flagellum. To illustrate this, when there are no mastigonemes on the front flagellum ($N_m = 0$) and the posterior flagellum is excluded ($f_2 = 0$, $L_2 = 0$), the anterior flagellum generates a thrust in negative $X$-direction ($U_X < 0$). This agrees well with previous studies modeling a smooth reciprocal beating flagellum (*Gray and Hancock, 1955*; *Hancock, 1953*; *Lauga and Powers, 2009*). But since $N_m > 2$, the velocity of the zoospore is reversed to positive $X$-direction due to the extra drag from the mastigonemes. This phenomenon was also described in hydrodynamics by *Namdeo et al., 2011*, and observed in experiment of *Cahill et al., 1996*. Interestingly, when changing directions, zoospores are observed to swim with the solely beating anterior flagellum to pull the body forwards while the posterior flagellum is immobile (*Figure 4(A)*) and *Video 5*, which confirms the thrust reversal ability of the mastigonemes. The beating frequency of the anterior flagellum in this case increases to $f_1 \approx 110\,\text{Hz}$. This finding also reassures the importance of mastigonemes on zoospore swimming, which is not similar to those of *C. reinhardtii* (*Amador et al., 2020*). The front flagellum of zoospores has fibrillar mastigonemes, similar to *Chlamydomonas*, but at higher density with tubular shape and larger in size, that could render into account of the different beating properties from the smooth posterior flagellum. However, high mastigoneme density (more than 20 per 1 μm flagellum length) shows mild effect on speed. From TEM images taken at the anterior flagellum, we estimate the mastigoneme density by averaging the number of mastigonemes manually counted over a flagellum length (See Appendix). We obtain $N_m = 13.0 \pm 0.8\,\mu\text{m}^{-1}$ (SD), which falls between the optimum range to generate speed (*Figure 3(D)*).

To understand how the coordination of two flagella influences zoospore speed, we vary the beating frequency of one flagellum while the other's remains constant and obtain the resultant speed (*Figure 3(E)*). We find that although both flagella contribute to zoospore speed, the anterior flagellum has larger impact on speed than the posterior does. For instance, the anterior flagellum can singly generate a speed threefold higher than the posterior flagellum can do at the same beating frequency. Moreover, the additional speed contributed by the anterior flagellum remains almost the same regardless of the variation of frequency of the posterior flagellum, while the speed contribution of the posterior flagellum decreases as the anterior flagellum increases its frequency. Since the contribution to zoospore speed is different between two flagella, we ask whether the energy consumption of each flagellum might also be different or equally distributed. In our model, each flagellar segment consumes a power deriving from the dot product between the drag force of water acting on the segment and the relative velocity of water to the segment, which can be written as $P \equiv \int \vec{v}_{w/f} \cdot d\vec{F}$ (see Appendix for detailed derivatives). We achieve that, the power consumption of each flagellum depends on both flagella at low frequencies ($f \leq 40\,\text{Hz}$), and becomes solely dependent on its own beating at high frequencies ($f \geq 40\,\text{Hz}$) (*Figure 3(F)*). At high frequencies, the anterior flagellum consumes approximately fivefold more power than the posterior, given the same beating frequency. As a result, the total power consumed for both flagella becomes less dependent on the posterior flagellum as the anterior flagellum beats faster (*Figure 3(G)*). In *Figure 3(H)*, we show the fraction of power consumed

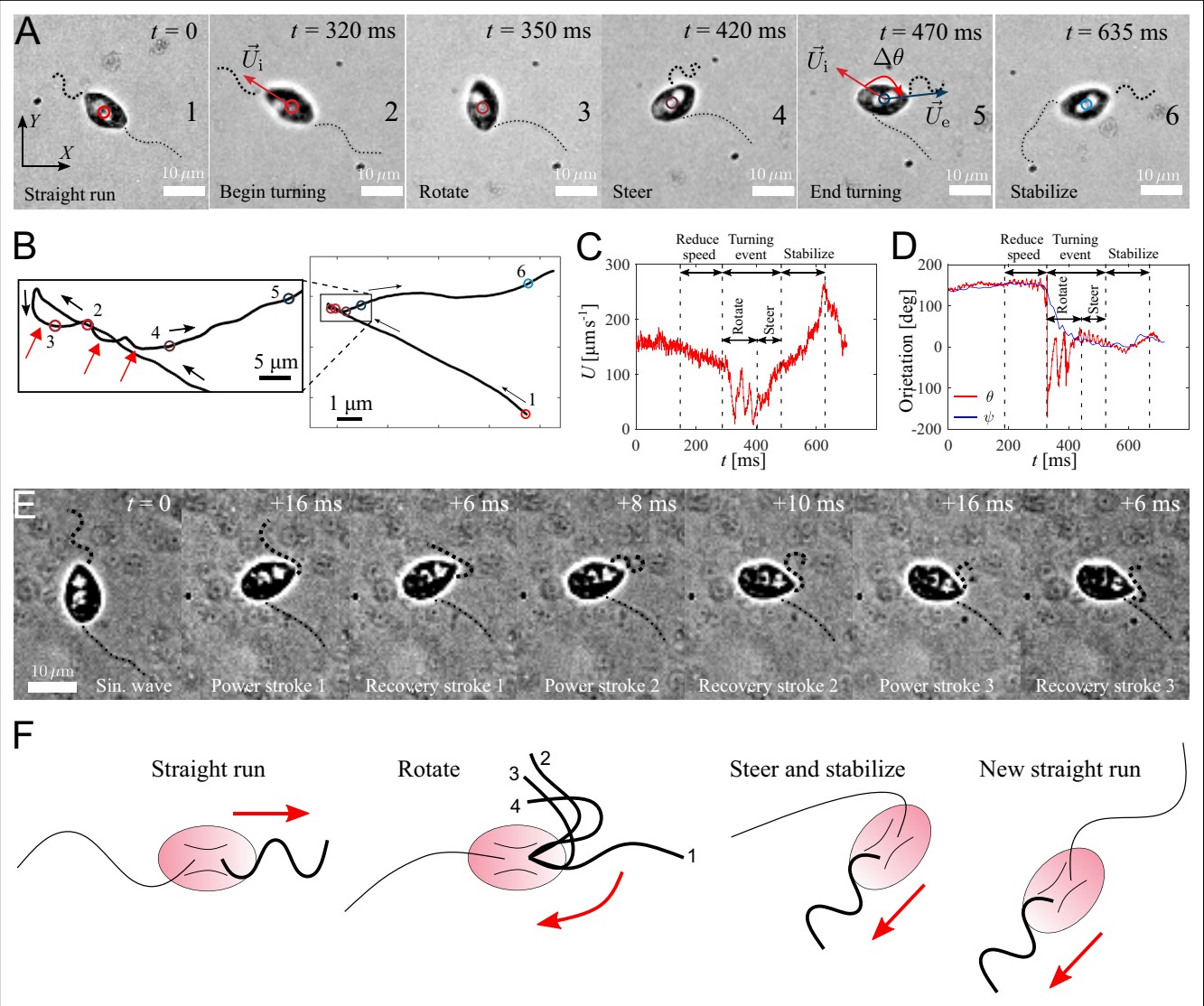

**Figure 4.** Active turning of individual *P. parasitica* zoospores in water. (**A**) Images of a zoospore changing direction. The two flagella cooperate to help the cell body rotate and steer to a new direction achieving a turning angle $\Delta\theta$. (**B**) Trajectory of the zoospore during the turning event. Three red arrows represent 3 back and forth stroke-like motions. (**C**) The speed $U$ of the zoospore during the turning event. The turning starts when the speed begins to fluctuate with large magnitude and lower frequency, and lasts for a duration of $\tau_s$ with a rotation of the cell body followed by steering to the new direction. (**D**) The moving directions $\theta$ and the body orientation $\psi$ of the zoospore during the turning event. (**E**) Images of the anterior flagellum of a zoospore beats with power and recovery stroke, similar to *C. reinhardtii*'s in a temporal zoom corresponding to the "Rotate" step of the turning event (but not from the same video as (**A**)). (**F**) Schematic to describe the gait of the flagella during a turning event. (1-2) Power stroke 1, (2-3) recovery stroke 1, (3-4) power stroke 2.

by the anterior flagellum over the total power consumption of the zoospore. We notice that, at the same beating frequency with the posterior flagellum, the anterior flagellum accounts for ~80 % of the total power. Interestingly, when the frequency of the posterior flagellum is ~1.7-fold higher than that of the anterior flagellum, both flagella consume the same amount of power. Indeed, this result agrees well with our experimental data, in which we obtain the anterior flagellum normally beats at 70 Hz and the posterior flagellum at 120 Hz. Thus, we can speculate that the energy is equally distributed for both flagella. In addition, we also estimate the propelling efficiency of the zoospore $\eta = \frac{P_0}{P_1+P_2}$, with $P_0$ as required power to move the cell body forward at speed $U_X$ (see Appendix for derivatives). We achieve that $\eta$ is higher as the anterior flagellum increases its frequency (*Figure 3(I)*). The efficiency reaches its maximum value at ~1.2 % when the beating frequency of the posterior is ~1.7-fold higher than that of the anterior flagellum. Overall, we show that the energy is shared in comparable

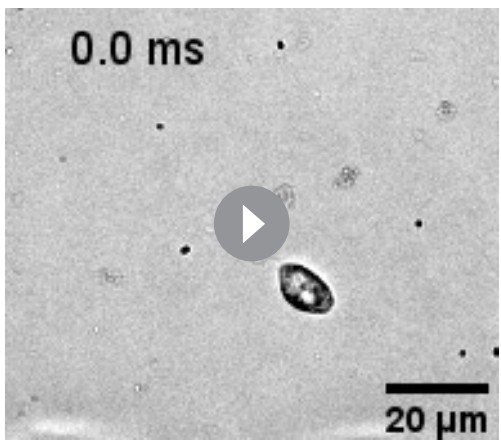

**Video 5.** A zoospore performing a turning event, which involves reducing speed, body rotation and steering resulting from active beating of anterior flagellum. The posterior flagellum is immobile during the turning event.

https://elifesciences.org/articles/71227/figures#video5

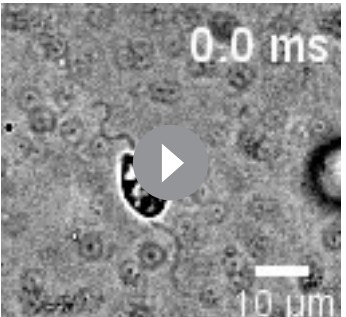

**Video 6.** A zoospore during a turning event changes gait of its anterior flagellum from sinuisoidal wave to power and recovery stroke, similar to *C. reinhardtii's*.

https://elifesciences.org/articles/71227/figures#video6

manner between two flagella, but the anterior flagellum has more influence on zoospore speed, power consumption and propelling efficiency. On the other hand, the posterior flagellum provides a modest contribution to zoospore speed despite beating at higher frequency and consuming half of the energy. Taking the fact that the anterior flagellum can beat singly with the posterior flagellum staying on pause during turning events, we speculate that the anterior flagellum is the main motor of zoospores. Nevertheless, the function of the posterior flagellum remains cryptic and we investigate more on its role during turning events.

## Turning events

We capture zoospores changing their directions by a unique active turning mechanism and an interesting coordination between two flagella. The videos of zoospore turning are recorded at 2000 fps with the same setup as the microscopic assay (*Figure 4(A)* and *Video 5*). Using Fiji and Trackmate, we track the positions of the zoospore and plot the trajectory, smoothed by moving average with step length $n = 40$, during its turning event (*Figure 4(B)*). We then compute the zoospore speed $U$ (*Figure 4(C)*), and the moving directions $\theta$ from the trajectory while manually measure body orientation $\psi$ over time (*Figure 4(D)*). We observe that at first, to prepare for the turn, the zoospore reduces its speed as both flagella beat with smaller amplitudes. Then, the posterior flagellum instantaneously stops beating, which marks the beginning of a turning event (*Figure 4* (A1-2)). The anterior flagellum now takes full control of zoospore motion during turning. The zoospore then perform two distinct sequences of motions right after the posterior stops beating: (i) rotation of the cell body out of the old direction resulting from a few repetitive stroke-like beating of the anterior flagellum (*Figure 4* (A3)), then (ii) steering towards a new direction as the anterior flagellum switches back to normal beating with sinusoidal waveform to propel the body (*Figure 4* (A4-5)). These two sequences of turning can be distinguished by two different patterns of trajectory during turning. While the rotation is indicated by multiple circular curves, in which each of them corresponds to a stroke-like back and forth beating motions, the steering results in straight line (*Figure 4(B)*). Additionally, the zoospore speed $U$ during this period also consists of two different patterns of fluctuations (*Figure 4(C)*), while the moving direction $\theta$ changes from large fluctuations (for rotation) to stable direction (for steering) (*Figure 4(C)*). The turning event ends when the cell body stably moves in the new direction, which is indicated by the recovery of speed and the overlap of the moving direction and body orientation. The anterior flagellum continues to propel the cell body out of the location of the turning event (~5 μm away), which we call 'stabilize' step (*Figure 4* (A6)). We notice that in this step, the anterior flagellum beats at a higher frequency than usual (~110 Hz, compared to normally at ~70 Hz) and achieves a high-speed of ~250 μm s⁻¹. Finally, the posterior flagellum resumes its beating and the zoospore returns to the normal straight run state.

It is striking that the anterior flagellum is able to completely change its gait from sinusoidal traveling wave to stroke-like beating during the rotation step of the turning event. Thus, we need to have experimental evidence from direct observation of the shape of the anterior flagellum during this gait changing period, which is sometimes hidden underneath the cell body due to the off-axis postion of the flagellar base, to extend our understanding on this turning behavior. We capture new videos of zoospore turning events, but the focus plane is slightly offset from the focus plane of the cell body (*Figure 4(E)* and *Video 6*). We observe that the anterior flagellum instantaneously perform continuous power and recovery strokes, similar to the breast-stroke beating of *Chlamydomonas*'s flagella. As a result, the cell body can rotate quickly about a fixed point, which helps the zoospore make a sharp turn. We summarize the turning of zoospores by a schematic in *Figure 4(F)*. Indeed, this rotation motion is similar to that of uniflagellate *C. reinhardtii* (*Bayly et al., 2011*), which strengthens the evidence for the actively switchable beating pattern of zoospores' anterior flagellum. While the anterior flagellum plays a major role in turning events on top of straight runs, the posterior flagellum only stays immobile throughout the process. Here, we can confirm that the posterior flagellum does not act as a rudder to steer the direction. Instead, it is fully stretched during turning events and might contribute to increase drag at one end of the cell body. The function of the posterior flagellum is not completely clear regarding thrust production during translation or turning. Thus, we speculate that it might contribute to other non-physical activities such as chemical and electrical sensing. The advantage of zoospore turning mechanism is that it allows them to actively and quickly achieve a new direction, which is not the case of other microswimmers such as the tumbling of *E. coli*'s (*Darnton et al., 2007*; *Perez Ipiña et al., 2019*) or *Chlamydomonas*'s (*Bennett and Golestanian, 2015*; *Polin et al., 2009*).

## Conclusion

We have performed the first systematic study of the swimming pattern and spreading features of *P. parasitica* zoospores, a plant pathogen, which is considered a major agricultural threat. Combining high-speed imaging and Resistive Force Theory, we show how the two opposite flagella are coordinated in producing thrust by beating together, allowing the microorganism to achieve high-speed swimming during straight runs. Furthermore, we find that turning is a coordinated process, in which the the posterior flagellum stops beating, while the anterior flagellum actively moves causing the cell body to rotate. Finally, we explain how fast-swimming periods and active turning events combine to produce a diffusion coefficient of $D = 3.5 \times 10^{-4}\,\mathrm{cm^2\,s^{-1}}$, a quantity that characterizes spatio-temporal spreading of this pathogen during plant epidemics.

It is worth stressing the motility pattern exhibited by the zoospores represents an Eukaryotic version of the 'run-and-tumble' motility class exhibited by bacteria peritrichous bacteria. Several eukaryotic swimmers, for example spermatozoa, do not exhibit such kind of motility pattern, but *C. reinhardtii* have been reported to also fall into this category (*Polin et al., 2009*). There are, however, important differences. *Chlamydomonas* possess two identical flagella located at one tip of the cell. Reorientation events occur during asynchronous beating periods of the two identical flagella, while straight runs require synchronous beating. In sharp contrast to this picture, we show that in zoospores while straight runs also involve a coordinated beating of the opposite and different flagella, turning involves temporary halting of posterior flagellum, while the anterior flagellum continues beating. This strongly suggests that *P. parasitica* navigates using a fundamentally different internal regulation mechanism to control swimming, than *C. reinhardtii*, a mechanism that is likely to be present in other Eukaryotic swimmers with two opposite and different flagella.

We believe our findings on the coordination of two flagella bring more insights on zoospore swimming dynamics. It was also not known from the literature about the role of each flagellum on the straight runs, and on turning events in particular. We show that although the energy is shared in comparable manner between both flagella, the anterior flagellum contributes more to zoospore speed. Zoospores actively change directions thanks to the sole beating of the anterior flagellum. We believe that anterior flagellum could be the main motor of zoospores that is in charge of generating speed, changing beating patterns from sinusoidal wave to power and recover stroke to quickly rotate the body, while the posterior flagellum might play a role in chemical/electrical sensing and providing an anchor-like turning point for zoospores, instead of acting like a rudder as previously hypothesized. These findings pave new ways for controlling the disease since now we can have different strategies on targeting one of the flagella.

## Materials and methods

### *P. parasitica* mycelium culture and zoospore release

We culture mycelium of *Phytophthora parasitica* (isolate 310, *Phytophthora* INRAE collection, Sophia-Antipolis, France) (*Bassani et al., 2020b*; *Galiana et al., 2019*) routinely on malt agar at 24 °C in the dark. To produce zoospores, we prepare the mycelium which is grown for one week in V8 liquid medium at 24 °C under continuous light. The material is then drained, macerated and incubated for a further four days on water agar (2 %) to induce sporangiogenesis. Zoospores are released from sporangia by a heat shock procedure. We place a petri-dish of mycelium inside a refrigerator at 4 °C for 30 min, then pour 10 mL of 37 °C distilled water on top of the mycelium and continue to incubate it at room temperature (25 °C) for another 30 min. Zoospores then escape from sporangia and swim up to the water. The zoospore suspension is then collected for further experiments.

### Scanning electron microscopy and transmission electron microscopy with negative staining

For Scanning Electron Microscopy and Transmission Electron Microscopy, cell pellets are fixed in a 2.5 % glutaraldehyde solution in 0.1 M sodium cacodylate buffer (pH 7.4) at room temperature (~25 °C) for 1 h and then stored at 4 °C. For Scanning EM observations, after three rinsing in distilled water, protists are filtered on a 0.2 μm isopore filter. Samples on filters are subsequently dehydrated in a series of ethanol baths (70 %, 96 %, 100 % three times, 15 min each). After a final bath in hexamethyldisilazane (HMDS, 5 min), samples are left to dry overnight. Samples on filters are mounted on Scanning EM stubs with silver paint and coated with platinum (3 nm) prior to observing. The Scanning EM observations are performed with a Jeol JSM-6700F scanning electron microscope at an accelerating voltage of 3 kV.

For TEM observations, samples are prepared using the negative staining method. After three rinsing in distilled water, a drop of cells suspension (~10 μL) is left for 5 min on a TEM copper grid (400 mesh) with a carbon support film. The excess liquid is removed with a filter paper. Subsequently, staining is done by adding a drop of 0.5 % (w/v) aqueous solution of uranyl acetate on the grid for 1.5 min, followed by removal of excess solution. The TEM observations are carried out with a JEOL JEM-1400 transmission electron microscope equipped with a Morada camera at 100 kV.

### Microscopic assays of zoospores

We pipette a droplet of 10 μL water containing zoospores onto a microscopic glass slide and spread the droplet to thoroughly cover the marked area of 1 × 1 cm. We then achieve a thin water film of approximately 100 μm thickness. We do not put coverslips on the water film to prevent the unwanted interactions between the zoospores and rigid surface of coverslips. We observe the swimming of individual zoospores inside the water film under a bright field transmission microscope (Nikon Eclipse T*i*2, Minato, Tokyo, Japan) at 40× objective with the high-speed camera Phantom v711 (Vision Research, NJ, USA). For the experiment to observe the swimming trajectories of the zoospores, we use 4× objective to capture a large swimming region of 5000 × 4000 μm. The captured images are processed by Fiji with Trackmate plugin.

### Estimation of trajectory parameters

Positions of an individual zoospore are captured at each time frame $\Delta t$. At each $t_j = j\Delta t$ ($j = 1, 2, 3, ...$), the zoospore has a position $z_j = (x(t_j), y(t_j))$. First, we smooth the trajectory by moving average with step $n$. The smoothed positions $Z_{j,n}$ are calculated as,

$$Z_{j,n} = \frac{1}{n} \sum_{j}^{n+j-1} z_j.$$

Each new position $Z_{j,n}$ possesses a velocity vector with speed,

$$U_{j,n} = \frac{1}{\Delta t} \| z_{j+n} - z_j \|,$$

and moving direction (angle between velocity vector and $x$-axis).

$$\theta_j = atan\left(\frac{y_{j+n} - y_j}{x_{j+n} - x_j}\right).$$

## Acknowledgements

This work has been supported by the French Government, managed by the National Research Agency under the Project UCA$^{JEDI}$ "Investissements d'Avenir Investments" bearing the reference n° ANR-15-IDEX-01. CCMA electron microscopy equipments have been funded by the Région Sud - Provence-Alpes-Côte d'Azur, the Conseil Général des Alpes Maritimes, and the GIS-IBiSA. We thank Ilaria Bassani, Nicolas Bruot, Xinhui Shen and Ngoc-Phu Tran for fruitful discussions, as well as Marcos, David Gonzalez-Rodriguez and the three reviewers for their careful reading and giving valuable feedback to improve the manuscript.

## Additional information

### Funding

| Funder | Grant reference number | Author |
| --- | --- | --- |
| French Government - Investissements d'Avenir - ANR | UCAJEDI ANR-15-IDEX-01 | Eric Galiana Xavier Noblin |

The funders had no role in study design, data collection and interpretation, or the decision to submit the work for publication.

### Author contributions

Quang D Tran, Conceptualization, Data curation, Formal analysis, Investigation, Methodology, Project administration, Software, Validation, Visualization, Writing – original draft, Writing – review and editing; Eric Galiana, Conceptualization, Funding acquisition, Methodology, Project administration, Resources, Writing – review and editing; Philippe Thomen, Conceptualization, Investigation, Methodology, Writing – review and editing; Céline Cohen, Conceptualization, Investigation, Writing – review and editing; François Orange, Investigation, Methodology, Writing – review and editing; Fernando Peruani, Formal analysis, Methodology, Software, Validation, Writing – review and editing; Xavier Noblin, Conceptualization, Data curation, Funding acquisition, Project administration, Supervision, Validation, Writing – review and editing

### Author ORCIDs

Quang D Tran http://orcid.org/0000-0002-5637-0647

### Decision letter and Author response

Decision letter https://doi.org/10.7554/eLife.71227.sa1
Author response https://doi.org/10.7554/eLife.71227.sa2

## Additional files

### Supplementary files

• Transparent reporting form

### Data availability

All data generated and simulation files are available via Zenodo using this URL: https://doi.org/10.5281/zenodo.4710633https://doi.org/10.5281/zenodo.4710633. In the data, we include: (1) datasets of all zoospore positions along multiple trajectories in the experiment of Figure 2, (2) a MATLAB file to compute all the statistical results in Figure 2(D-G), (3) a MATLAB file containing the simulation model presented in Figure 2(H), (4) datasets of zoospore positions, speed, moving directions, body orientations during the turning, presented in Figure 4(A-D).

The following dataset was generated:

| Author(s) | Year | Dataset title | Dataset URL | Database and Identifier |
|---|---|---|---|---|
| Tran QD, Galiana E, Thomen P, Cohen C, Orange F, Peruani F, Noblin X | 2021 | Coordination of two opposite flagella allows high-speed swimming and active turning of individual zoospores | https://doi.org/10.5281/zenodo.4710633 | Zenodo, 10.5281/zenodo.4710633 |

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

## Appendix 1

### Obtaining characteristics of beating flagella by kymograph

We first use Fiji with StackReg plugin to fix the position of the cell body in the swimming videos, and only the two flagella remain beating.

We retrieve the parameters of the beating flagella by applying kymographs on multiple cross-sections which are along to or normal to the moving direction at the positions shown in the figure below. These kymographs help converting the complex time-dependent 2D image data of the flagella from $(X, Y, t)$ to separate signals $(X, t)$ and $(Y, t)$. As a result, the kymographs at cross-section (1) and (2) give us the average amplitudes $A$ of the waveform shapes of the two flagella, and the ones at (3) and (4) provide us with information of beating frequencies $f$ and wavelengths $\lambda$.

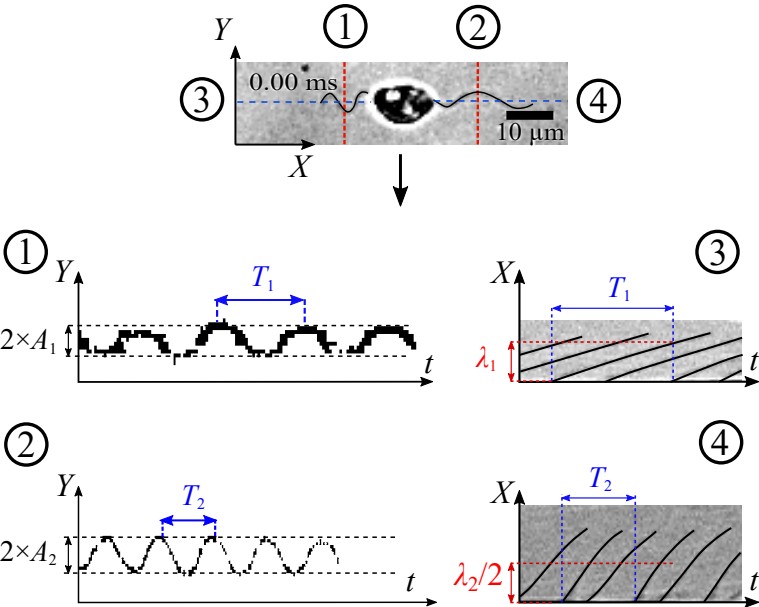

**Appendix 1—figure 1.** Strategy to apply kymograph to obtain characteristics of beating zoospore flagella.

## Appendix 2

### Variation of $U_{th}$

To verify if the way we chose the value of $U_{th}$ will affect the result of turning angles and other swimming parameters, we vary the $U_{th}$ value by ±10 % of the chosen value, which ranges from 100 to 122.5 μm s⁻¹. We plot the running time $\tau_r$, stopping time $\tau_s$ and turning angles $\Delta\theta$ according to the varied $U_{th}$. Data are shown in the figure below. We see that the changes do not affect the results of $\tau_r$, $\tau_s$ and $\Delta\theta$ estimation. Thus, we believe the sensitivity of $U_{th}$ selection can be tolerated by ±10%.

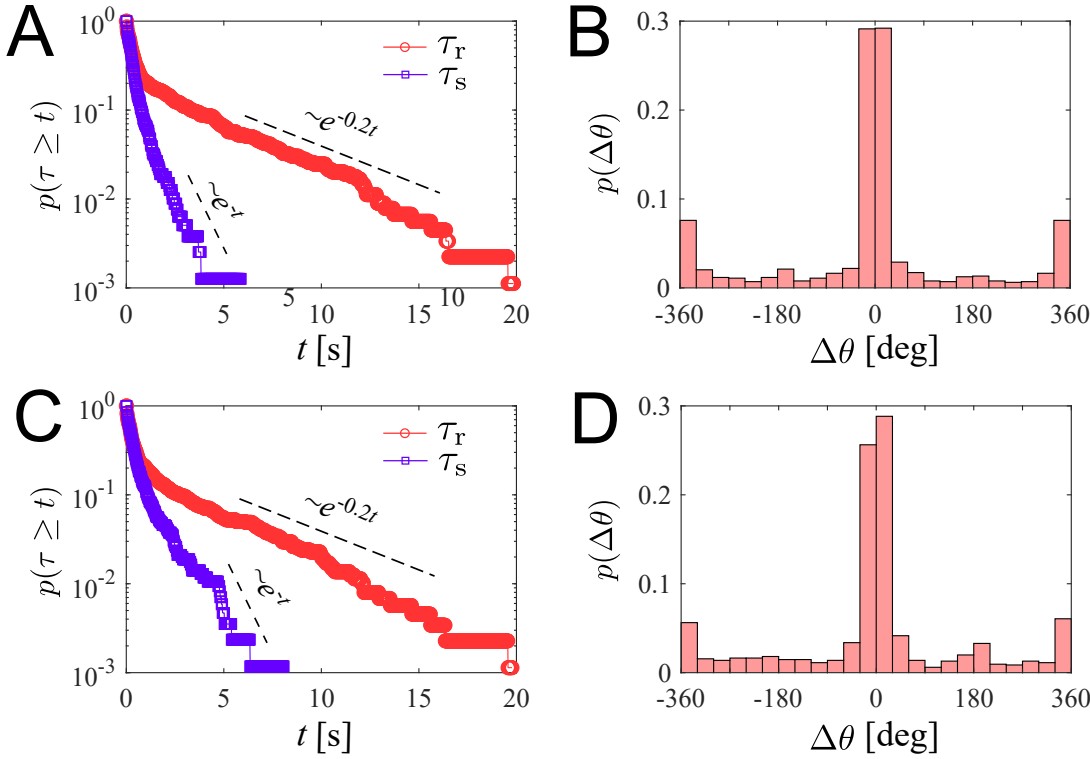

**Appendix 2—figure 1.** Estimation of $\tau_r$, $\tau_s$ and $\Delta\theta$ for different variation of $U_{th}$. (**A**) $\tau_r$, $\tau_s$ and (**B**) $\Delta\theta$ for $U_{th} - 10\%$ (**C**), $\tau_r$ $\tau_s$ and (**D**) $\Delta\theta$ for $U_{th} + 10\%$.

## Appendix 3

### Characteristics of helical trajectories

From the microscopic assays with high-speed camera and 40× objective, we observe zoospores swimming in helical trajectories. We manually measure the characteristics of these helical trajectories and present the data in the table below.

**Appendix 3—table 1.** Characteristics of helical trajectories of zoospores during straight runs.

| Sample | Pitch $p$ ($\mu$m) | Radius $R$ ($\mu$m) | Gyrational speed $\dot{\phi}$ (rad s$^{-1}$) |
|---|---|---|---|
| 1 | 122 | 3.250 | 3.64 $\pi$ |
| 2 | 123 | 4.435 | 4.00 $\pi$ |
| 3 | 145 | 4.129 | 2.86 $\pi$ |
| 4 | 164 | 3.390 | 2.5 $\pi$ |
| 5 | 178 | 4.850 | 2.1 $\pi$ |
| 6 | 112 | 3.010 | 4.44 $\pi$ |
| 7 | 120 | 4.300 | 4.44 $\pi$ |
| 8 | 96 | 4.244 | 5 $\pi$ |
| 9 | 127 | 4.090 | 3.33 $\pi$ |
| 10 | 106 | 4.744 | 4 $\pi$ |

## Appendix 4

### Estimation of mastigoneme density

Mastigoneme density is an important parameter, which determines the thrust reversal effect of the anterior flagellum and generates speed for zoospores. Using TEM, we capture images of anterior flagellum containing the mastigonemes. We first select a longitudinal flagellum length and manually count the total number of mastigonemes attached along that distance (See figure below). The mastigoneme density $N_m$ is calculated as the ratio between the number of counted mastigonemes and the selected flagellum length.

**Appendix 4—table 1.** Measurement data of mastigonemes in TEM images.

| Sample | No. of mastigonemes | Flagellum length ($\mu$m) | Density ($\mu$m$^{-1}$) |
|---|---|---|---|
| 1 | 152 | 11.842 | 12.8357 |
| 2 | 168 | 11.597 | 14.4865 |
| 3 | 113 | 8.627 | 13.0984 |
| 4 | 181 | 13.590 | 13.3186 |
| 5 | 198 | 15.195 | 13.0306 |
| 6 | 177 | 13.994 | 12.6483 |
| 7 | 88 | 7.499 | 11.7349 |

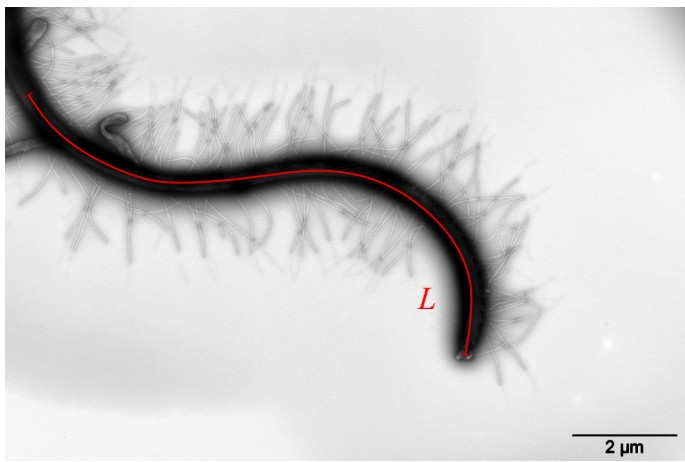

**Appendix 4—figure 1.** Strategy to estimate mastigoneme density.

## Appendix 5

### Hydrodynamics of two flagella in a swimming zoospore using Resistive Force Theory

Following the schematics of *Figure 3(a)* in the main text, we consider a parameter $s_k$ as the distance along the flagellum from the root to the segment $ds_k$, then we define dimensionless constants $\alpha_k$ as the ratios of the wavelength to the arc length of the anterior ($k = 1$) and posterior flagellum ($k = 2$), respectively. Whereas

$$\alpha_1 = \frac{\lambda_1}{L_1} = \frac{x_1 - c}{s_1} \tag{6}$$

and

$$\alpha_2 = \frac{\lambda_2}{L_2} = \frac{-(x_2 + c)}{s_2}. \tag{7}$$

Thus, we can express the segment position $\vec{r}_k$ in terms of $s_k$, as follows

$$\vec{r}_1 = (\alpha_1 s_1 + c)\vec{i} + A_1 \sin\left(\omega_1 t - \frac{2\pi s_1}{L_1}\right)\vec{j}, \tag{8}$$

and

$$\vec{r}_2 = (-\alpha_2 s_2 - c)\vec{i} + A_2 \sin\left(\omega_2 t - \frac{2\pi s_2}{L_2}\right)\vec{j}. \tag{9}$$

We achieve the velocity of the flagella relative to the cell body as

$$\vec{v}_{f/b} = d\vec{r}_k/dt = \omega_k A_k \cos\left(\omega_k t - \frac{2\pi s_k}{L_k}\right)\vec{j}. \tag{10}$$

The beating of flagella results in the movement of the cell body relative to water, with a velocity:

$$\vec{v}_{b/w} = U_X \vec{i}. \tag{11}$$

We neglect the velocity of the cell body in $Y$-direction with the assumption that the zoospore swimming is very directional and $U_X >> U_Y$.

Resistive Force Theory (RFT) states that the drag force by fluid acting on an infinitesimal segment $ds$ of the flagellum is proportional to the relative velocity of fluid to the flagellum segment, as follows

$$\frac{d\vec{F}}{ds} = K_N V_N \vec{n} + K_L V_L \vec{l}, \tag{12}$$

where $V_N$ and $V_L$ are two components of relative velocity of fluid in normal and tangent direction to the flagellum segment, $K_N$ and $K_L$ are the drag coefficients of the flagellum in normal and tangent to the flagellum segment, $\hat{n}$ and $\hat{l}$ are the unit vectors normal and tangent to the flagellum segment, respectively.

*Equation (12)* can be expressed as:

$$\begin{aligned}
\frac{d\vec{F}}{ds} &= K_N(\vec{v}_{w/f} \cdot \vec{n})\vec{n} + K_L(\vec{v}_{w/f} \cdot \vec{l})\vec{l} \\
&= K_N\left(\vec{v}_{w/f} - (\vec{v}_{w/f} \cdot \vec{l})\vec{l}\right) + K_L(\vec{v}_{w/f} \cdot \vec{l})\vec{l} \\
&= K_N\left[(\vec{v}_{w/f} \cdot \vec{l})\left(\frac{K_L}{K_N} - 1\right)\vec{l} + \vec{v}_{w/f}\right],
\end{aligned} \tag{13}$$

where $\vec{v}_{w/f}$ is the relative velocity of water to the flagellum, which is achieved as

$$\vec{v}_{w/f} = -\vec{v}_{b/w} - \vec{v}_{f/b}, \tag{14}$$

and the tangent unit vector $\hat{l}$ is expressed as

$$\vec{l} = \frac{1}{|\frac{\partial \vec{r}}{\partial s}|}\frac{\partial \vec{r}}{\partial s}. \tag{15}$$

During the derivative, we use an assumption that the flagella have small amplitude deflection ($A << L$). Thus, we can approximate that $\cos^2\left(\frac{2\pi s_k}{L_k} - \omega_k t\right) \approx 0.5$. This helps to simplify the term $\left|\frac{\partial \vec{r}}{\partial s}\right| = \sqrt{\alpha_k^2 + 4\pi^2 \frac{A_k^2}{L_k^2} \cos^2\left(\frac{2\pi s_k}{L_k} - \omega_k t\right)} = \sqrt{\alpha_k^2 + 2\pi^2 \frac{A_k^2}{L_k^2}}$.

The drag coefficients $K_N$ and $K_L$ is estimated by Brennen and Winet, as follows:

$$K_{\mathrm{N}} = \frac{4\pi\mu}{\ln\left(\frac{4\lambda}{d}\right) - 2.90}, \tag{16}$$

$$K_{\mathrm{L}} = \frac{2\pi\mu}{\ln\left(\frac{4\lambda}{d}\right) - 1.90}, \tag{17}$$

where $\mu = 8.9 \times 10^{-4}\,\mathrm{Pa\,s}$ is the water viscosity at $25\,^{\circ}\mathrm{C}$, $\lambda$ is the wavelength and $d$ is the diameter of the flagellum. With $U_Y$ being neglected, we also consider that the two flagella do not generate force in $Y$-direction, but only induce thrust in $X$-direction. Hence,

$$\mathrm{d}\vec{F} \approx \mathrm{d}F_X \vec{i} = (\mathrm{d}\vec{F} \cdot \vec{i})\vec{i}. \tag{18}$$

We then apply RFT on each flagellum of the zoospore to calculate the total drag force acting on it. For the posterior flagellum, each segment $\mathrm{d}s_2$ is a simple smooth and slender filament, having drag coefficients $K_{\mathrm{N2}}$ and $K_{\mathrm{L2}}$. following *equation (16) and (17)*. From *equation (13) and (18)*, we derive the drag force of water acting on posterior flagellum as

$$F_{X,2} = K_{\mathrm{N2}}L_2\left[\frac{-2\pi^2 v_{\mathrm{w2}}(\gamma_2 - 1)\beta_2^2 - (\gamma_2 - 1)U_X}{1 + 2\pi^2\beta_2^2} - U_X\right], \tag{19}$$

where $v_{\mathrm{w2}} = \lambda_2 f_2$ is the wave propagation velocity, $\gamma_2 = K_{\mathrm{L2}}/K_{\mathrm{N2}}$, $\beta_2 = A_2/\lambda_2$. For the anterior flagellum, each segment $\mathrm{d}s_1$ contains additional $N\mathrm{d}s_1$ mastigonemes which are considered perpendicular to the segment itself. These mastigonemes also act as slender filaments experienced drag from water. Interestingly, due to the direction arrangement, the relative velocity normal to the flagellum segment results in drag force in tangent direction to the mastigonemes, and subsequently, the relative velocity tangent to the flagellum segment results in drag force in normal direction to the mastigonemes. Hence, the total drag force acting on a segment of the anterior flagellum is derived as:

$$\frac{\mathrm{d}\vec{F}_1}{\mathrm{d}s_1} = (K_{\mathrm{Nf1}} + NhK_{\mathrm{Lm1}})V_{\mathrm{N1}}\vec{n}_1 + (K_{\mathrm{Lf1}} + NhK_{\mathrm{Nm1}})V_{\mathrm{L1}}\vec{l}_1, \tag{20}$$

where $N$ is the density of mastigonemes, $h$ is the length of each mastigoneme, $V_{\mathrm{N1}}$ and $V_{\mathrm{L1}}$ are two components of relative velocity of fluid in normal and tangent direction to the segment $\mathrm{d}s_1$, $K_{\mathrm{Nf1}}$ and $K_{\mathrm{Lf1}}$ are the drag coefficients in normal and tangent direction of the flagellum filament, respectively; $K_{\mathrm{Nm1}}$ and $K_{\mathrm{Lm1}}$ are the drag coefficients in normal and tangent direction of the mastigonemes, respectively. $K_{\mathrm{Nf1}}$, $K_{\mathrm{Lf1}}$, $K_{\mathrm{Nm1}}$ and $K_{\mathrm{Lm1}}$ are also calculated from *equation (16) and (17)*. In another perspective, we can consider the anterior flagellum receives additional drag from the mastigonemes, which is presented by two increased drag coefficients in normal and tangent direction defined as

$$K_{\mathrm{N1}} = (K_{\mathrm{Nf1}} + N_{\mathrm{m}}hK_{\mathrm{Lm1}}) \tag{21}$$

and

$$K_{\mathrm{L1}} = (K_{\mathrm{Lf1}} + N_{\mathrm{m}}hK_{\mathrm{Nm1}}), \tag{22}$$

respectively. Here, $N_{\mathrm{m}}$ is the density of mastigonemes; $h$ is the length of each mastigoneme; $K_{\mathrm{Nf1}}$ and $K_{\mathrm{Lf1}}$ are the drag coefficients in normal and tangent direction of the flagellum filament, respectively; $K_{\mathrm{Nm1}}$ and $K_{\mathrm{Lm1}}$ are the drag coefficients in normal and tangent direction of the mastigonemes, respectively. $K_{\mathrm{Nf1}}$, $K_{\mathrm{Lf1}}$, $K_{\mathrm{Nm1}}$ and $K_{\mathrm{Lm1}}$ are also calculated from *equation (16) and (17)*. The total fluid drag force acting on the anterior flagellum is also derived from *equation 13 and 18*:

$$F_{X,1} = K_{\mathrm{N1}}L_1\left[\frac{2\pi^2 v_{\mathrm{w1}}(\gamma_1 - 1)\beta_1^2 - (\gamma_1 - 1)U_X}{1 + 2\pi^2\beta_1^2} - U_X\right], \tag{23}$$

where $v_{w1} = \lambda_1 f_1$ is the wave propagation velocity of the anterior flagellum, $\gamma_1 = K_{L1}/K_{N1}$, $\beta_1 = A_1/\lambda_1$.

At the same time, the ellipsoidal cell body moving with velocity $U_X$ also experiences a drag force from water (following Happel and Brenner)

$$\vec{F}_{d,cell} = -6\pi\mu b\xi_e U_X \vec{i}, \tag{24}$$

where $\mu$ is water viscosity, $\xi_e$ is the shape coefficient of the ellipse cell body in 2D. $\xi_e$ is estimated as

$$\xi_e = \frac{4/3\left(\kappa^2 - 1\right)}{\frac{2\kappa^2 - 1}{\sqrt{\kappa^2 - 1}}\ln\left(\kappa + \sqrt{\kappa^2 - 1}\right) - \kappa}, \tag{25}$$

with $\kappa = a/b$ being the ellipse body ratio. In low Reynolds number condition, total forces equate to zero due to approximately zero inertia. Thus,

$$\Sigma\vec{F} = \vec{F}_1 + \vec{F}_2 + \vec{F}_{d,cell} = \vec{0}. \tag{26}$$

Combine **equation (19), (23) and (24)**, we achieve translational velocity $U_X$ of the zoospore as shown in **Equation 5**.

## Appendix 6

### Power and efficiency of two flagella during in zoospore swimming

We derive the power that each flagellum generates during zoospore swimming to investigate how the two flagella cooperate and contribute to the swimming. The power generated by the anterior flagellum $P_1$ is calculated as

$$P_1 = \int_{s_1=0}^{L_1} \vec{v}_1 \cdot d\vec{F}_1, \tag{27}$$

where $\vec{v}_1 = \vec{v}_{w/f(1)}$ is the relative velocity vector of water to each flagellar segment, and $d\vec{F}_1$ is the drag force of water acting on each flagellar segment. From (13), we have

$$\vec{v}_1 \cdot d\vec{F}_1 = K_{N1} \left[ (\gamma_1 - 1) \left( \vec{v}_1 \cdot \hat{l}_1 \right)^2 + \vec{v}_1^2 \right] ds_1. \tag{28}$$

Then, we can derive $P_1$ from (14), (15) and (28) as

$$P_1 = K_{N1} L_1 \left[ (\gamma_1 - 1) \frac{(2\pi^2 v_1 \beta_1^2 - U_X)^2}{1 + 2\pi^2 \beta_1^2} + U_X^2 + 2\pi^2 v_{w1}^2 \beta_1^2 \right]. \tag{29}$$

Similarly, we can also derive $P_2$ as

$$P_2 = K_{N2} L_2 \left[ (\gamma_2 - 1) \frac{(2\pi^2 v_2 \beta_2^2 + U_X)^2}{1 + 2\pi^2 \beta_2^2} + U_X^2 + 2\pi^2 v_{w2}^2 \beta_2^2 \right]. \tag{30}$$

The useful power $P_0$ required to propel the cell body with a speed $U_X$ can be derived as

$$P_0 = F_{d,cell} \cdot U_X = 6\pi \mu b \xi_e U_X^2. \tag{31}$$

The efficiency of the two flagella in propelling the cell body is estimated as

$$\eta = \frac{P_0}{P_1 + P_2}. \tag{32}$$

## Appendix 7

### Dimensions and physical parameters of zoospore's flagella

With the characteristics of beating flagella obtained by kymograph, we calculate the values of drag coefficients $K_N$, $K_L$, flagellum shape ratio $\beta$, and drag coefficient ratio $\gamma$. The results are presented in the table below.

**Appendix 7—table 1.** Physical parameters of beating flagella of *P parasitica* zoospores in normal conditions.

| Structures | Parameters | Denotes | Value |
|---|---|---|---|
| Anterior | Amplitude ($\mu$m) | $A_1$ | 1.6 |
| flagellum | Wavelength ($\mu$m) | $\lambda_1$ | 7.5 |
| | Length ($\mu$m) | $L_1$ | 23.4 |
| | Beating freq. (Hz) | $f_1$ | 74 |
| | Drag coeff. of filaments | $K_{Nf1}$ | $6.5589 \cdot 10^{-15}$ |
| | (N s $\mu$m$^{-2}$) | | |
| | | $K_{Lf1}$ | $2.067 \cdot 10^{-15}$ |
| | Drag coeff. of mastigonemes | $K_{Nm}$ | $2.7906 \cdot 10^{-15}$ |
| | (N s $\mu$m$^{-2}$) | | |
| | | $K_{Lm}$ | $1.1167 \cdot 10^{-15}$ |
| | Mastigoneme length ($\mu$m) | $h$ | 1.5 |
| | Mastigoneme density ($\mu$m$^{-1}$) | $N_m$ | 13 |
| | ($A_1/\lambda_1$)ratio | $\beta_1$ | 0.21 |
| | Drag coeff. ratio ($K_{L1}/K_{N1}$) | $\gamma_1$ | 2.047 |
| Posterior | Amplitude ($\mu$m) | $A_2$ | 2.9 |
| flagellum | Wavelength ($\mu$m) | $\lambda_2$ | 15.17 |
| | Beating freq. (Hz) | $f_2$ | 128 |
| | Length ($\mu$m) | $L_2$ | 25.8 |
| | Drag coefficients | $K_{N2}$ | $4.6415 \cdot 10^{-15}$ |
| | (N s $\mu$m$^{-2}$) | | |
| | | $K_{L2}$ | $1.6401 \cdot 10^{-15}$ |
| | ($A_2/\lambda_2$)ratio | $\beta_2$ | 0.19 |
| | Drag coeff. ratio ($K_{L2}/K_{N2}$) | $\gamma_2$ | 0.3534 |

## Appendix 8

### Data publication

Our experimental data are accessible via Zenodo (https://doi.org/10.5281/zenodo.4710633).

In the data, we include

1. Datasets of all zoospore positions along multiple trajectories in the experiment of *Figure 2*,
2. A MATLAB file to compute all the statistical results in *Figure 2(D–G)*,
3. A MATLAB file containing the simulation model presented in *Figure 2(H)*,
4. Datasets of zoospore positions, speed, moving directions, body orientations during the turning, presented in *Figure 4(A–D)*.

