## [Editor Report]

The authors present a study of the swimming behaviour of the zoospores of the water mold *Phytophthora* (Greek "Plant Destroyer"), which is responsible for significant crop damage worldwide. The motility of the zoospores is likely a significant contributor to the successful spread of the disease, and as such its study has potential wide impact. The authors suggest using a model that the anterior "hairy" (covered with mastigonemes) flagellum is the primary contributor to motility, and show with high-speed imaging that the microorganism is able to turn on the spot by stopping its posterior flagellum, and changing the beat-pattern of its anterior flagellum from "sperm-like" to "*Chlamydomonas*-like".

---

## [Decision Letter]

**Decision letter after peer review:**

Thank you for sending your article entitled "Cooperation of two opposite flagella allows high-speed swimming and active turning in zoospores" for peer review at *eLife*. Your article is being evaluated by 2 peer reviewers, and the evaluation is being overseen by a Reviewing Editor and Aleksandra Walczak as the Senior Editor.

While the reviewers and editor are supportive of the paper, it was our consensus opinion that a substantial amount of revision to the manuscript is needed to clarify the points below. Furthermore, since active turning and steering feature prominently in the title and abstract, it would be crucial to further investigate the flagellar motion and gait changes during turns. Without this, we believe that the conclusions of the paper are overstated.

*Reviewer #2:*

The capacity for swimming microorganisms to efficiently explore their environment can be essential for their survival and proliferation. This manuscript by Galiana et al., examines the swimming characteristics of P. parasitica, a plant pathogen known for causing widespread disease and economic loss. The authors present high-speed video recordings which reveal how individual cells execute turning events during their swimming motion. There is considerable experimental data, which in combination with a mathematical model for flagellar motion, sheds light on how the two flagella collectively produce locomotion and turning events.

The turning events are largely portrayed as being due to temporary arresting of the posterior flagellar beating. E.g., "turning requires the posterior flagellum to halt, while the anterior one continue beating". However, while stopping the posterior beating is necessary, it is not sufficient. What is lacking is a detailed explanation of precisely how the cells turn. From Video 4, it appears that the anterior flagellum changes its gait completely from a sinusoidal travelling wave to something more reminiscent of *Chlamydomonas*, with distinct power and recovery strokes (but with mastigonemes). The oscillations of angle of Figure 4d indeed suggest that a distinct gait is employed to turn the cell. The authors mention this on line 389, but this is not investigated further, citing imaging capabilities. Cell turning events seem to require coordinating the stop/start motion of one flagellum and distinct gait changes in the other, but the mechanism(s) behind this remain unclear.

From the trajectories presented in Figure 2, numerous quantities are extracted, including swimming speed, turning angle, run times, etc. The swimming speed threshold is essential for delineating turns events from 'runs', and one method for extracting U_th_ is presented. This leads to the conclusion that the most likely reorientation angle (Δ_θ_, see Figure 2g) is actually 0 degrees. i.e. the cell stops and then continues in the same direction. It could be that this is an artefact of a relatively high value of U_th_, which misclassifies one run into two smaller ones. This sensitivity is further shown in Figure 2b – increasing U_th_ to just ~125 microns/second would result in one single turning event (tau_s_) instead of three distinct ones. It is not clear from the data how the results would change if U_th_ were varied in a sensitivity analysis, or defined in another way.

The authors have presented an experimental system which is capable of recording reasonably long trajectories of swimming P. parasitica cells, and an analysis pipeline which can extract statistical distributions of various swimming properties. The overarching assumption is that the 2D measurements can be extrapolated to three dimensions. Given the displayed trajectories are quite long (several millimetres), it seems that there are very few issues with the cells swimming out of focus. It is likely that the shallow chamber (100 microns) helps to maintain the cells within the focal plane and helps to minimise biases in the run time measurement. However, if that is the case, it's hard to know the effect that confinement has on the motility characteristics. It could be that the stop-start motion with Δ_θ_=0 actually represents failed out-of-plane turning events.

The authors utilised microscopic parameters from the swimming motion to predict the bulk diffusivity. It was mentioned that the direct measurement of D is unreliable given the small number of trajectories and their short duration. However, this seems to be inconsistent with the supplementary videos, which show cells in focus for extended periods of time and allow collection of trajectory data. Even if individual MSD measurements cannot be collected in large sets, perhaps a bulk diffusivity could be measured at lower magnification. At the moment, there is no strong quantitative link between the simulated diffusivity and the actual spreading of cells.

A simplified mathematical model is developed in which resistive force theory (RFT) applied to the two flagella calculates the propulsive force on the cell body. This is then used to derive a closed form expression for the swimming speed (Equation 5), which predicts values commensurate with the observed swimming speeds (Figure 3). It is an interesting finding that the mastigonemes modify the sign of the propulsive force (also noted in previous works), enabling the anterior flagellum to pull the cell through the fluid. The planar motion of the flagellum in the model seems reasonable given the dynamics presented in Video 3. However, it is unclear by what mechanism the cells rotate as they swim. Perhaps this has something to do with the off-axis position of the flagellar base. Since the helical motion of the cells features prominently in the paper and is likely important for overall dispersal, it would be great if the model could account for viscous torques, calculate the cell body rotation rate and characteristics of the helical path.

The observed dynamics have been classified into straight runs and turning events. However, many of the 'runs' are certainly not straight (see Figure 2). It is not clear whether these curved trajectories are due to rotational diffusion, or perhaps from hydrodynamic interactions with the boundaries (bacteria are known to swim in circles near no-slip walls). This could have significant implications for the dispersal calculations, and for classification of events within the trajectories.

In order to calculate a cell's 'run time', it is necessary to visualise two successive turning events which bookend a single run. However, for cells swimming in three dimensions, the probability of observing a given run length depends on the value of the run length. Can the authors please provide information about any biases in their data collection, and the extent of any biases in the presented distributions? This could include discussion of depth of field, chamber height, and whether any cells were excluded.

The motility strategy examined in this paper is described as "steering" and is presented as a possible method for pathogenic cells to target the plant host. However, the dynamics are all spatially isotropic, and it is not clear how cells could navigate towards specific chemical cues (e.g. potassium, physical cues, electrical signals).

The motor efficiency is determined in terms of the swimming speed to wave propagation. There are many ways of calculating efficiency in low Reynolds number hydrodynamics, many of which include power/energy calculations. The authors allude to a link between power and drag due to mastigonemes, but it's hard to determine from the presented results how much energy the cell needs to swim. I think it would be very helpful to expand the section on efficiency, connecting with other established methods, and results for other organisms.

Quantifying the microscale trajectory features and predicting the collective cell diffusivity requires a threshold value of U_th_. It would be very important to examine the sensitivity of the measurements to the value of U_th_, and explore the implications if another definition was chosen.

Figure 2. Some of the time-dependent processes are plotted in terms of frame number (t/dt), and others in absolute time (t). It would be best to include all processes in terms of t, so that conclusions are independent of frame rate.

Line 38 mentions that *E. coli* possess "a passive helical flagellum", but they possess a bundle of these. Please correct the numbers.

There is a need to better distinguish between axial rotation and cell "rotation" during turns. Perhaps it would be better not to use the same word.

Line 58. The organisms can apparently swim for several hours. However, this may be very short compared to the timescales required to find a plant. Can the authors please elaborate on how motility can be sustained in a typical environment, and how long it might take for a cell to reach its target (assuming no active navigation)?

The article should be carefully read by a native English speaker. For example, "underwhelming knowledge" is not an appropriate way of describing open questions. "Flimsy" also refers to an object's structural integrity and not its shape. The word "accumulating" is used extensively throughout the paper but is not appropriate since it implies temporal integrating. i.e drag and velocity don't 'accumulate' along the flagellum.

*Reviewer #3 (Recommendations for the authors):*

The authors present a study of the swimming behaviour of the zoospores of the water mold Phytopthora, which combines experimental imaging of fixed cells (to examine structure), image capture and analysis of multiple swimming cells to examine bulk swimming behaviour, high-speed imaging to characterise on the spot turning behaviour, and an analytical mathematical model of the straight "run" swimming behaviour that incorporates the mastigonemes on the anterior flagellum.

In terms of what is already known about these systems, it is already known that Phytophora zoospores swim with two flagella at the high speeds shown herein, and that the anterior flagellum is covered with mastigonemes (see for instance the authors' previous work DOI: 10.1016/j.csbj.2020.10.045). It has also been known for some time that mastigonemes reverse the expected direction of propulsion for a flagellum beating with a sperm-like (travelling wave) waveform, so this behaviour is expected, and indeed commented on in the authors' previous work. The model provided is based upon previous models (see eg https://doi.org/10.1063/1.3608240), which should be made clearer in the text, with the adaptation being that there are 2 flagella and a body here.

The main novel components are (1) the discussion that the anterior flagellum accounts for the majority of the propulsion, and (2) the characterisation of the on-the-spot turning (as far as I know). The first point is based on analysis from the analytical model. My concerns with this conclusion are three-fold. Firstly, it relies on mastigonemes remaining rigid and normal to the tangent of the main flagellum, whereas some bending with the flow is to be expected (reducing their impact). Secondly, the resistive force theory model, while qualitatively powerful, does not include hydrodynamic interactions between neighbouring mastigonemes, which may also reduce their impact. Thirdly, it is quite sensitive to the mastigoneme density once this drops to around 10 per micron (the way in which the mastigoneme density was estimated, and the data, should be made clear in the manuscript), and fourthly, the posterior flagellum naturally beats at around twice the frequency (probably because they are expending the same amount of energy), so I do not feel that the conclusion "we consider the anterior flagellum as the main motor of zoospores because it has the ability to immensely increase or decrease speed with a small adjustment of its beating frequency" is justified.

The characterisation of the on-the spot turning from high speed image microscopy is very nice, to the best of my knowledge novel, and the analogues to peritrichous bacteria and *Chlamydomonas* well drawn up. This section is well described (though not modelled), but would probably benefit from a schematic in addition to the set of experimental images.

The paper would benefit from a clearer exposition of what the novel components are in relation to previous work, and a link back to the motivation of crop damage and how this understanding of motility may be used in future pest control strategies.

While I enjoyed reading the article, the authors should endeavour to make it clearer in certain places what has come before, and what is the novel component. For instance, the modelling seems to follow the arguments of https://doi.org/10.1063/1.3608240, and while this is cited elsewhere, I believe this should be stated very clearer in the model section. Secondly, there are a few sentences which mislead with respect to aspects of the novelty, for instance:

1. "Despite the relevance of zoospore spreading in the epidemics of plant diseases, it is not known how these zoospores swim and steer with two opposite beating flagella." For the swimming this is clearly known, as the authors commented on in their previous work, it is the turning I believe that was not well characterised.

2. "Here, we introduce a new type of microswimmer, named Phytophthora zoospores, which has two different flagella collaborating for unique swimming and turning mechanisms (Figure 1(a))." – you have presented work previously on the swimming of this organism.

I believe the data on mastigoneme density is critical to one of your conclusions, and should be presented carefully, and I believe this conclusion about the anterior flagellum being the main driver for motility is probably not correct, more broadly – it certainly needs more careful treatment.

---

## [Author Response]

While the reviewers and editor are supportive of the paper, it was our consensus opinion that a substantial amount of revision to the manuscript is needed to clarify the points below. Furthermore, since active turning and steering feature prominently in the title and abstract, it would be crucial to further investigate the flagellar motion and gait changes during turns. Without this, we believe that the conclusions of the paper are overstated.

We have performed additional experiments to observe the shape of the zoospore flagella during turning. Indeed, as the Reviewer 2 predicted, the anterior flagellum is shown to perform power and recovery strokes, similarly to *Chlamydomonas*’s. It also shows that the anterior flagellum of zoospores can completely change its gait from sinusoidal beating to distinct power and recovery strokes during turning events. We believe the experimental evidence is more convincing than the simulation as we planned in our Action plan. So, we instead just performed this experiment and did not add the simulation into our article. The details of the new experiment are presented in the question below of Reviewer 2. Additionally, we have changed the title of the paper into “Coordination of two opposite flagella allows high-speed swimming and active turning of individual zoospores”, as we think coordination is a more suitable word than “cooperation” to describe the functions of the two flagella. Also, in the conclusion, line 466, we have changed the term “turning requires halting of posterior flagellum…” into “ turning involves temporary halting of posterior flagellum…” The word “requires” might be misleading since it is probably not a condition for the turning.

Reviewer #2 (Recommendations for the authors):The capacity for swimming microorganisms to efficiently explore their environment can be essential for their survival and proliferation. This manuscript by Galiana et al., examines the swimming characteristics of P. parasitica, a plant pathogen known for causing widespread disease and economic loss. The authors present high-speed video recordings which reveal how individual cells execute turning events during their swimming motion. There is considerable experimental data, which in combination with a mathematical model for flagellar motion, sheds light on how the two flagella collectively produce locomotion and turning events.The turning events are largely portrayed as being due to temporary arresting of the posterior flagellar beating. E.g., "turning requires the posterior flagellum to halt, while the anterior one continue beating". However, while stopping the posterior beating is necessary, it is not sufficient. What is lacking is a detailed explanation of precisely how the cells turn. From Video 4, it appears that the anterior flagellum changes its gait completely from a sinusoidal travelling wave to something more reminiscent of Chlamydomonas, with distinct power and recovery strokes (but with mastigonemes). The oscillations of angle of Figure 4d indeed suggest that a distinct gait is employed to turn the cell. The authors mention this on line 389, but this is not investigated further, citing imaging capabilities. Cell turning events seem to require coordinating the stop/start motion of one flagellum and distinct gait changes in the other, but the mechanism(s) behind this remain unclear.

We thank the Reviewer for a very important suggestion that the turning mechanism of zoospores shares similarities with the behavior observed in *Chlamydomonas*. Indeed, Figure 4D suggests that turning in zoospores occurs via power and recovery strokes. Despite lacking of 3D observations of the flagella during turning, we observe the zoospore turning at different 2D view angles and notice that the turning happens on a same focus plane. This suggests that the anterior flagellum has changed its swimming pattern but its beating is still a planar motion. Importantly, we observe that during turning the dynamics of the anterior flagellum is fundamentally different from the sinusoidal waveform observed during run phases. Moreover, we find that the anterior flagellum of zoospores, which possesses mastigonemes, performs power and recovery strokes, similarly to the flagellum beating of *Chlamydomonas*. We also observed that while the anterior flagellum takes control of the zoospores during turning, the posterior flagellum completely stops beating. We speculate that the posterior flagellum contributes to chemical/electrical sensing.

Though a full fluid mechanics understanding of the turning – which is beyond the scope of the current manuscript – is missing, our study identifies key distinct elements of the turning of zoospores: beyond providing a statistical characterization of the duration and turning angles of these events, we observe that during turning the posterior flagellum does not beat, while the anterior one does it, and its gait is different from the sinusoidal travelling wave that characterizes run phases. These observations suggest that turning in zoospore is different from turning observed in *Chlamydomonas* and other microorganisms. We stress that to the best of our knowledge, this is the first systematic quantitative study of the swimming pattern of *Phytophthora*. A detailed fluid mechanics understanding of the turning is beyond the scope of the current study and represents an experimental and theoretical challenge.

We have added Figure 4E and Supp. Video 6 to show the gait changing of the anterior flagellum during turning, and Figure 4F to show the schematics summarizing the turning event of zoospores. We have rewritten the whole sub-section “Turning events” (from line 399 to 447) to better describe this gait changing mechanism. We also include the gait changing finding in the abstract (line 2427):

“Furthermore, we find that turning involves a complex active process, in which the posterior flagellum temporarily stops, while the anterior flagellum keeps on beating and changes its pattern from sinusoidal waves to power and recovery strokes, similar to *Chlamydomonas*’s breaststroke, to reorient its body to a new direction”

From the trajectories presented in Figure 2, numerous quantities are extracted, including swimming speed, turning angle, run times, etc. The swimming speed threshold is essential for delineating turns events from 'runs', and one method for extracting U_th_ is presented. This leads to the conclusion that the most likely reorientation angle (Δ_θ_, see Figure 2g) is actually 0 degrees. i.e. the cell stops and then continues in the same direction. It could be that this is an artefact of a relatively high value of U_th_, which misclassifies one run into two smaller ones. This sensitivity is further shown in Figure 2b – increasing U_th_ to just ~125 microns/second would result in one single turning event (tau_s_) instead of three distinct ones. It is not clear from the data how the results would change if U_th_ were varied in a sensitivity analysis, or defined in another way.

To verify if the way we chose the value of *U*_th_ will affect results of turning angles, run and turning events durations, we have varied the *U*_th_ value by ±10% of the chosen value, ranging from 100 to 122.5 μm s^-1^. We plot the running time, stopping time and turning angles according to the varied *U*_th_ and add the data in the Appendix 2. We see that the changes do not affect much the estimations of average tau_s_ and tau_r_ or the statistics of Δ*θ*. Thus, we believe the sensitivity of *U*_th_ selection can be tolerated by 10%.

We have added a description for this sensitivity of *U*_th_ from line 192-194 (Section “Statistics of individual swimming patterns”):

“The sensitivity of our *U*_th_ selection can be tolerated by ±10 % of the chosen value, ranging from 100 to 122.5 μm s^-1^ (See Appendix)”

The authors have presented an experimental system which is capable of recording reasonably long trajectories of swimming P. parasitica cells, and an analysis pipeline which can extract statistical distributions of various swimming properties. The overarching assumption is that the 2D measurements can be extrapolated to three dimensions. Given the displayed trajectories are quite long (several millimetres), it seems that there are very few issues with the cells swimming out of focus. It is likely that the shallow chamber (100 microns) helps to maintain the cells within the focal plane and helps to minimise biases in the run time measurement. However, if that is the case, it's hard to know the effect that confinement has on the motility characteristics. It could be that the stop-start motion with Δ_θ_=0 actually represents failed out-of-plane turning events.

As zoospores experience aerotaxis, they naturally swim close to the water surface. Thus, we design our experimental system as a swimming pool, instead of a chamber, because we do not cover the top of the sample droplet to avoid unwanted physical interaction between zoospores and the cover. To produce this “swimming pool”, we first draw a square of 1cm × 1cm on a glass slide, then pipette a droplet of 10μl of solution with zoospores into that square. After that, we gently spread the droplet to cover all the surface area of the square and achieved a thin layer of liquid with height of ~100μm. Though we have introduced this method in the experimental section at the end of the manuscript, we think that the setup needs to be described more clearly in the main texts. We have added a few sentences to better introduce this setup (line 160-162, section “Statistics of individual swimming patterns”):

“The setup of the water thin film can be visualized as a “swimming pool” that is not covered as we want to avoid the unwanted physical interactions of zoospores with the top when they experience aerotaxis…”

Moreover, we agree with the Reviewer that the dominant Δ*θ* = 0 could result from the failed out-of-plane movement close to the water surface. We also observe this behavior experimentally. Thus we have added Video 2 and implemented this speculation into the discussion of the Δ*θ* result, as follows (line 206-208):

“It is also shown that zoospores preferentially turn with the angle around 0^o^, which we speculate that it results from the failed out-of-plane movement when zoospores swim near the water surface during their aerotaxis (see Video 2).”

The authors utilised microscopic parameters from the swimming motion to predict the bulk diffusivity. It was mentioned that the direct measurement of D is unreliable given the small number of trajectories and their short duration. However, this seems to be inconsistent with the supplementary videos, which show cells in focus for extended periods of time and allow collection of trajectory data. Even if individual MSD measurements cannot be collected in large sets, perhaps a bulk diffusivity could be measured at lower magnification. At the moment, there is no strong quantitative link between the simulated diffusivity and the actual spreading of cells.

This was indeed done as can be seen in the inset of Figure 2I (for the MSD, from which *D* can be extracted): *D* = 3.5 × 10^-4^ cm^2^ s^-1^. However, we note that the average duration of runs is 5 seconds and the observed trajectories are typically less than 40 seconds in length, and so involving not more than 8 runs on average. Furthermore, the directions of motion of consecutive run phases are correlated (see Figure 2G). Estimations of the diffusion coefficient based on direct measurements of individual trajectories are highly unreliable (in general, and particularly for the experimental limitations of the experiment). However, the collected data allows a reliable estimate of the statistics of run phases and of the correlations between consecutive run phases. These quantities fully determine the large-scale properties of the diffusive process and provide the most reliable way to estimate the diffusion coefficient even when a much larger number of longer trajectories is available. As a proof of this, see the dispersion in MSD in simulations where we used 1h trajectories. Reliable estimations on MSD measurements require a large number of long trajectories, which are typically out of reach experimentally (for instance, direct measurements of MSD has repeatedly lead to wrongly report anomalous transport, etc).

Low magnification experiments that track the expansion of an initial high concentration drop of microorganisms provide an alternative reliable method, but here again, there is no direct measurement of the MSD of individuals, but based on the assumption of an underlying diffusion process where the diffusion coefficient *D* is a fitting parameter. In an initial high concentration, however, interactions among microorganisms can lead to a series of collective mechanisms that impact the behavior of the microorganisms and estimation of the diffusion process.

In summary, we are convinced that the performed analysis is the most reliable method to estimate the diffusion coefficient of individual zoospores and insist that the diffusion coefficient is fully determined – as in any random walk process – by the properties and correlations of run phases (as well as duration of turning angle).

We have added a few sentences to further discuss this point (line 230-234):

“We emphasize that *D* represents the estimation of diffusion coefficient of individual swimming of zoospores from random walk process. This is more to show the intrinsic ability of individual zoospores to perform spatial exploration, rather than to quantify the bulk diffusivity where the collective swimming behaviors, which involve zoospore-zoospore interactions, play a major role.”

A simplified mathematical model is developed in which resistive force theory (RFT) applied to the two flagella calculates the propulsive force on the cell body. This is then used to derive a closed form expression for the swimming speed (Equation 5), which predicts values commensurate with the observed swimming speeds (Figure 3). It is an interesting finding that the mastigonemes modify the sign of the propulsive force (also noted in previous works), enabling the anterior flagellum to pull the cell through the fluid. The planar motion of the flagellum in the model seems reasonable given the dynamics presented in Video 3. However, it is unclear by what mechanism the cells rotate as they swim. Perhaps this has something to do with the off-axis position of the flagellar base. Since the helical motion of the cells features prominently in the paper and is likely important for overall dispersal, it would be great if the model could account for viscous torques, calculate the cell body rotation rate and characteristics of the helical path.

We agree with the reviewer that the body self-rotation might result from the off-axis position of the flagellar base. However, we believe positions of the flagella and the kidney-shaped cell body might contribute to the chirality of zoospores, which causes the spontaneous self-rotation together with the translation and results in a helical trajectory (10.1140/epjst/e2016-60054-6). Although we have implemented this explanation in the section “Straight runs”, it was not very clearly stated and the estimation of the body rotation rate and characteristics of the helical path were not well-presented. We have revised this part, as follows (line 248-256):

“While translating, the cell body gyrates around the moving direction simultaneously, which results in a helical swimming trajectory (see Supp. Video 4 for a long run of a zoospore swimming in water). We believe that this gyrational motion might result from the intrinsic chiral shape of the zoospore body and off-axis arrangement of their flagella (Figure 1(B)). Indeed, previous studies have shown that chirality of a microswimmer’s body induces spontaneous axial rotation resulting from the translational motion (*Keaveny and Shelley, 2009*; *Namdeo et al., 2014*; *Löwen, 2016*). From multiple observations, we obtained the pitch and radius of the helical trajectories at *p* = 130 ± 8 µm (SEM) and *R* = 4.0 ± 0*.*2 µm (SEM), respectively (data presented in Appendix). We then estimate the gyrational speed of the cell body ϕ = 2π /Δ_t_, where Δ_t_ is the duration the zoospore travels through a full turn of the helical path (Figure 3(B)). We obtain ϕ = (3.6 ± 0.3)π rad s^-1^ (SEM) (see Appendix).”

The reviewer suggested revising the model to account the viscous torques. However, we believe this body self-rotation (or we call “gyration” to distinguish with the rotation during turning) does not contribute to the translational motion. The two flagella, during reciprocal beating, can generate small vibrations in axial direction of the cell body, but not the directional axial rotation. Thus, the gyration of the cell body solely depends on the chirality of the zoospores, which can be considered as a passive motion resulting from the translation. Integrating the chirality of the zoospores into the model to describe the gyration is out of scope of the paper since we are focusing on the thrust generation of the two flagella. However, we have instead obtained the gyration rate and characteristics of the helical paths from experimental observations.

We have added some explanation, as follows (line 272-275):

“We assume that the gyration of the cell body does not affect the shapes and motions of the flagella since the beating frequencies of the two flagella are much higher than the gyrational speed. The gyration also does not contribute to the translation as we consider it as a passive motion resulting from the chirality of zoospores. Thus, the swimming zoospore can be considered as a 2D model…”

The observed dynamics have been classified into straight runs and turning events. However, many of the 'runs' are certainly not straight (see Figure 2). It is not clear whether these curved trajectories are due to rotational diffusion, or perhaps from hydrodynamic interactions with the boundaries (bacteria are known to swim in circles near no-slip walls). This could have significant implications for the dispersal calculations, and for classification of events within the trajectories.

Certainly, bacteria as well as zoospores are subject to thermal fluctuations, and thus mathematical straight trajectories do not exist. Abrupt changes in trajectories are systematically related to a turning event. We used the term “straight” to indicate that run phases do not display neither abrupt changes of direction neither a finite curvature (as when motion is chiral). Note that the anterior and posterior flagellum do not rotates as the flagellar bundle of *E. coli* does, see Figure 3. Thus, there is no reason to expect that a no-slip wall will induce a hydrodynamic torque as occurs in *E. coli*. Furthermore, we do not observe chiral motion near the bottom surface. Furthermore, note that for bacteria, oriented curvature associated to trajectories (on a given surface), exhibits the same sign: i.e., it is expected that bacteria on the surface turn in the same direction. It is evident that this does not occur for zoospores.

We have added some sentences to discuss this point (line 256-261):

“The observations of helical trajectories also confirm that each flagellum of zoospores beat as a flexible oar in a 2D plane as we observe the two flagella flattened into two straight lines during the gyration. Thus, zoospores are not expected to swim in circles when interacting with no-slip boundaries as seen in case of *E. coli* with a rotating flagellar bundle. The “curved straight runs” of zoospores that we observed in Figure 2(A) might result from rotational diffusion and thermal fluctuations.”

In order to calculate a cell's 'run time', it is necessary to visualise two successive turning events which bookend a single run. However, for cells swimming in three dimensions, the probability of observing a given run length depends on the value of the run length. Can the authors please provide information about any biases in their data collection, and the extent of any biases in the presented distributions? This could include discussion of depth of field, chamber height, and whether any cells were excluded.

The height of the thin film of water containing zoospores is ~100μm, while the speed of zoospores is averaged at 150μm/s, we assume the swimming of zoospores in our “swimming pool” is straight as interactions with substrate will result in disruption of the straight runs. We believe the zoospores can swim straight in a long time due to their aerotaxis, which makes them tend to stay close to the water surface. We have discussed this aerotaxis in the section of “Statistics of individual swimming”.

We used survival curves to obtain the statistics of running time and stopping time because we were aware of that there were situations the swimmers cross the observation region without making a turn. Thus, we obtained the probability that the running time or stopping time is greater than a time period.

Except for the dead cells that keep fluctuating around certain positions during the whole video, we did not exclude any swimmers from the data.

The motility strategy examined in this paper is described as "steering" and is presented as a possible method for pathogenic cells to target the plant host. However, the dynamics are all spatially isotropic, and it is not clear how cells could navigate towards specific chemical cues (e.g. potassium, physical cues, electrical signals).

It is known that zoospore can perform chemotaxis (see https://doi.org/10.1016/j.fbr.2007.02.001, https://doi.org/10.1038/nrmicro1064 or in our recent review https://10.1016/j.csbj.2020.10.045). In shorts, near the plant-root, zoospores can perceive diverse stimuli at multiple levels. The ion exchange dynamics between soil particles and plant roots, together with the chemical gradients generated by root exudates, can activate cell responses. This results in biased motion towards a preferential localization at the interface between soil particles and plant roots. However, it is not known how zoospore steer towards a desired direction, or how flagella dynamics is affected in the presence of chemical gradients.

We have added this description in the introduction part, as follows (line 66-73):

“Previous studies have shown that during the spreading and approaching the host, zoospores can have complicated swimming patterns and behaviors as they experience multiple interactions with environmental signals, both physical, electrical and chemical, in soil and host-root surface (Bassani et al., 2020a). Near the plant-root, zoospores can perceive various stimuli from the environment, such as ion exchange between soil particles and plant roots, the chemical gradients generated by root exudates, which activate cell responses. This results in coordinated behaviors of zoospores, allowing them to preferentially navigate to the water film at the interface between soil particles and plant roots.”

The motor efficiency is determined in terms of the swimming speed to wave propagation. There are many ways of calculating efficiency in low Reynolds number hydrodynamics, many of which include power/energy calculations. The authors allude to a link between power and drag due to mastigonemes, but it's hard to determine from the presented results how much energy the cell needs to swim. I think it would be very helpful to expand the section on efficiency, connecting with other established methods, and results for other organisms.

We thank the Reviewer for this suggestion. We have calculated the power consumption of each flagellum as well as the propelling efficiency of the zoospore. This question is similar to the question of Reviewer 3. Please find the detail answer in the response below for Reviewer 3.

Quantifying the microscale trajectory features and predicting the collective cell diffusivity requires a threshold value of U_th_. It would be very important to examine the sensitivity of the measurements to the value of U_th_, and explore the implications if another definition was chosen.

We have varied the U_th_ value and show that 10% or slightly higher variation of its value does not affect results. See our reply to this point above.

Figure 2. Some of the time-dependent processes are plotted in terms of frame number (t/dt), and others in absolute time (t). It would be best to include all processes in terms of t, so that conclusions are independent of frame rate.

We have already fixed the Figure 2 according to reviewer’s suggestion.

Line 38 mentions that E. coli possess "a passive helical flagellum", but they possess a bundle of these. Please correct the numbers.

We have fixed the number. The new sentence is now:

“*Escherichia coli* is one of the most studied prokaryotic swimmers, which possesses a bundle of passive helical flagella…”

There is a need to better distinguish between axial rotation and cell "rotation" during turns. Perhaps it would be better not to use the same word.

We name this axial rotation as “gyration”. The cell rotation during turning is remained as “rotation”. We have revised the texts accordingly.

Line 58. The organisms can apparently swim for several hours. However, this may be very short compared to the timescales required to find a plant. Can the authors please elaborate on how motility can be sustained in a typical environment, and how long it might take for a cell to reach its target (assuming no active navigation)?

In natural ecosystems and even more in agro-systems, putative host plants are close.

This “proximity’ makes the distance to find a plant relatively short and it is compatible to timeability of zoospore to swim. Moreover, zoospores can sense the electrical and chemical signals released from the plant roots, which helps them navigate the direction towards the target.

We have added a few sentences to elaborate this point (line 57-60):

“To facilitate the spreading, their cell bodies store an amount of energy (mycolaminarin, lipid) allowing them to swim continuously for several hours (Judelson and Blanco, 2005). In natural ecosystems and even more in agro-systems, putative host plants are usually close. This proximity makes the distance to find a plant relatively short and it is compatible to the time-ability of zoospores to swim”

The article should be carefully read by a native English speaker. For example, "underwhelming knowledge" is not an appropriate way of describing open questions. "Flimsy" also refers to an object's structural integrity and not its shape. The word "accumulating" is used extensively throughout the paper but is not appropriate since it implies temporal integrating. i.e drag and velocity don't 'accumulate' along the flagellum.

We have rewritten the whole sentence with the term “underwhelming knowledge”.

The word “flimsy” has been removed, and the word “accumulating” has been replaced by suitable words.

Reviewer #3 (Recommendations for the authors):The authors present a study of the swimming behaviour of the zoospores of the water mold Phytopthora, which combines experimental imaging of fixed cells (to examine structure), image capture and analysis of multiple swimming cells to examine bulk swimming behaviour, high-speed imaging to characterise on the spot turning behaviour, and an analytical mathematical model of the straight "run" swimming behaviour that incorporates the mastigonemes on the anterior flagellum.In terms of what is already known about these systems, it is already known that Phytophora zoospores swim with two flagella at the high speeds shown herein, and that the anterior flagellum is covered with mastigonemes (see for instance the authors' previous work DOI: 10.1016/j.csbj.2020.10.045). It has also been known for some time that mastigonemes reverse the expected direction of propulsion for a flagellum beating with a sperm-like (travelling wave) waveform, so this behaviour is expected, and indeed commented on in the authors' previous work. The model provided is based upon previous models (see eg https://doi.org/10.1063/1.3608240), which should be made clearer in the text, with the adaptation being that there are 2 flagella and a body here.

We thank Reviewer 3 for the comment. As the Reviewer stated, it is already known that *Phytophthora* zoospores swim with two sinusoidal waveform flagella, and the mastigonemes reverse the thrust. In our previous review (DOI: 10.1016/j.csbj.2020.10.045), we have recapped the current knowledge about zoospore swimming, which involves the anterior flagellum with mastigonemes and smooth posterior flagellum both beating to propel the body forwards. However, the characteristics of the beating flagella during swimming such as beating frequencies, amplitudes, wavelengths (for example, the posterior flagellum beats with frequency 1.7-fold higher than the anterior flagellum), together with characteristics of the zoospore swimming trajectories such as body lateral rotation speed, turning angles, running time, turning time, are not known.

Moreover, our work also looks at individual contribution of each flagellum on zoospore swimming, and how the two flagella are coordinated during swimming and turning. We develop a model based on Resistive Force Theory (RFT), consisting of both flagella and a cell body, to better understand that cooperation. To construct this model, we have consulted from many previous studies using RFT, including the paper from Namdeo *et al.*, as the Reviewer has mentioned (https://doi.org/10.1063/1.3608240). Though we have cited the paper when confirming the thrust reversal effect in zoospores, we agree with the Reviewer that it needs to be stated more clearly. Following the Reviewer’s recommendations, we have made the reference of the previous model from Namdeo *et al.,* clearer and emphasize our scopes better. Below are the changes made:

Abstract, line 19-21:

“Despite the relevance of zoospore spreading in the epidemics of plant diseases, characteristics of individual swimming of zoospores have not been fully investigated. It remains unknown about the characteristics of two opposite beating flagella during translation and turning, and the roles of each flagellum on zoospore swimming…”

Introduction section, line 93-99:

“Although it has been known about how zoospores swim, characteristics of the swimming and the beating flagella have not been statistically reported. The effects of mastigonemes on zoospore swimming also need to be carefully investigated since the mechanical properties of mastigonemes such as size, rigidity, density, can affect the swimming differently (*Namdeo et al., 2011*). For instance, while mastigonemes are shown to generate thrust reversal in *P. palmivora* zoospores (*Cahill et al., 1996*), they do not contribute to enhance swimming of *C. reinhardtii* (*Amador et al., 2020*).”

Introduction section, line 117-126:

“ We obtain statistics of the trajectories and develop a numerical model to study and extrapolate the zoospore spreading characteristics solely by random walks. Then, we detail an in-depth study on the hydrodynamics of *P. parasitica*’s flagella and acquire a mathematical model to correlate the functions of two flagella on the motion of straight runs. Although theoretical models for microswimmers with single mastigonemes-attached flagella have been formulated (*Brennen, 1975; Namdeo et al., 2011*), models for microswimmers with two heterokont flagella have yet been considered as in case of zoospores. Here, we use Resistive Force Theory and further develop the model of a single flagellum with mastigonemes (*Brennen, 1975; Namdeo et al., 2011*) to adapt it with another smooth flagellum and a cell body, using a hypothesis of no interactions between two flagella…”

The main novel components are (1) the discussion that the anterior flagellum accounts for the majority of the propulsion, and (2) the characterisation of the on-the-spot turning (as far as I know). The first point is based on analysis from the analytical model. My concerns with this conclusion are three-fold. Firstly, it relies on mastigonemes remaining rigid and normal to the tangent of the main flagellum, whereas some bending with the flow is to be expected (reducing their impact).

We agree with the Reviewer that the thrust reversal effect of the mastigonemes depends on their flexibility. In fact, this flexibility has been carefully characterized in the paper of Namdeo et al., (https://doi.org/10.1063/1.3608240) and Khaderi et al.,

(https://doi.org/10.1103/PhysRevE.79.046304). The flexibility of mastigonemes can be defined by a dimensionless parameter *F*_m_ = 12*µKAωh*^3^/(*Ed*_m_^3^) where *µ* is the fluid viscosity, *K* = 2*π*/*λ*, *A* is the amplitude of the beating flagellum, *ω* = 2*πf, h* is the height of mastigonemes, *d*_m_ is the diameter of mastigonemes, and *E* is the Young modulus of mastigonemes. *F*_m_ is the ratio between viscous force and elastic force on mastigonemes, thus can be used to characterize the flexibility of the mastigonemes. With *F*_m_ < 0.1, mastigonemes are considered as rigid and the thrust reversal effect is preserved.

In our case, the zoospores’ mastigonemes have *F*_m_ at order of 10^-4^, which is much lower than 0.1. Thus, we can assume that the mastigonemes of zoospores are rigid and non-deformable. We realize that we did not clarify our assumption well enough.

Thus, we have added the following information for the flexibility of the mastigonemes (subsection Straight runs, line 283-294):

“Additionally, there are multiple tubular type-1 mastigonemes with length *h* and diameter *d*_m_ attached to the surface of the anterior flagellum with density *N*_m_ indicating the number of mastigonemes attached on a unit length of the flagellum. It is important to determine the flexibility of these mastigonemes as it would impact the ability of the mastigonemes to produce drag. We estimate the flexibility by a dimensionless parameter, which were carefully characterized in previous studies (*Namdeo et al., 2011*; *Khaderi et al., 2009*), *F*_m_ = 12*µKAωh*^3^/(*Ed*_m_^3^), where *µ* is the fluid viscosity, *K* = 2*π*/*λ* is the wave number, *E* is the Young modulus of the mastigonemes. With this estimation, if *F*_m_
*<* 0.1, mastigonemes are considered as fully rigid. In case of zoospores’ mastigonemes, we achieve *F*_m_ at order of 10^-4^, which is much lower than 0.1. Thus, we can assume that the mastigonemes of zoospores are non-deformable and rigidly attached to the anterior flagellum. As a result, hydrodynamic interactions between neighboring mastigonemes can also be neglected.”

Secondly, the resistive force theory model, while qualitatively powerful, does not include hydrodynamic interactions between neighbouring mastigonemes, which may also reduce their impact.

Since we now have the flexibility parameter *F*_m_ defined, our zoospore mastigonemes can be treated as fully rigid and non-deformable. Thus, we can neglect the effect of hydrodynamic interactions between neighboring mastigonemes.

Thirdly, it is quite sensitive to the mastigoneme density once this drops to around 10 per micron (the way in which the mastigoneme density was estimated, and the data, should be made clear in the manuscript),

We have re-analyze the mastigoneme density following the Reviewer’s suggestion. We have also introduced the method and experimental data to estimate the mastigoneme density in the Appendix 4. In short, the mastigoneme density *N*_m_ is defined as the number of mastigonemes attached to a unit length of the anterior flagellum. The number of mastigonemes were manually counted, then divided by the flagellum length that contains all those counted mastigonemes. The measurement data show that *N*_m_ = 13.0 ± 0.8 µm^-1^ (SD).

and fourthly, the posterior flagellum naturally beats at around twice the frequency (probably because they are expending the same amount of energy), so I do not feel that the conclusion "we consider the anterior flagellum as the main motor of zoospores because it has the ability to immensely increase or decrease speed with a small adjustment of its beating frequency" is justified.

The Reviewer has raised a very interesting point that both flagella might expend the same amount of energy, which leads to the posterior flagellum beats at twice the frequency as that of the anterior flagellum. The conclusion we made about the anterior flagellum as the main motor indeed needs a better explanation. To further investigate this, we have calculated the power generated by each beating flagellum to propel the cell body forward. In our model, each flagellar segment moves in water and received a drag force from water. Hence, the power produced by a flagellar segment results from the dot product between the drag force of water acting on the segment and the relative velocity of water to the segment, which can be written as dP≡v→w/f⋅dF→. From this equation, we can derive the power generated by each flagellum *P*_1_ and *P*_2_, and estimate the propelling efficiency of the two flagella as *η* = *P*_0_/(*P*_1_ + *P*_2_), where *P*_0_ = *F*_d_ . *U*_*x*_ is the useful power required to propel the cell body forward with a speed *U_x_*. The detail derivative is added to the Appendix 6. Then, we plotted 4 new figures, 3F, G, H, I to show the correlation of the power of each flagellum, the propelling efficiency of two flagella according to different beating frequencies of the two flagella.

We have re-written the whole paragraph to correctly conclude about the cooperation of the two flagella, from line 362 to 398 of subsection “Straight runs”.

The characterisation of the on-the spot turning from high speed image microscopy is very nice, to the best of my knowledge novel, and the analogues to peritrichous bacteria and Chlamydomonas well drawn up. This section is well described (though not modelled), but would probably benefit from a schematic in addition to the set of experimental images.

We thank the reviewer for the compliment. We have added the schematics showing the shape of the two flagella during the turning event in Figure 4F.

The paper would benefit from a clearer exposition of what the novel components are in relation to previous work, and a link back to the motivation of crop damage and how this understanding of motility may be used in future pest control strategies.

We have added a discussion to explain clearer the novel components in our study and link back to the motivation of crop damage to emphasize the importance of this study which can be used in pest control strategies. The new discussion is as follows (line 470-480):

“We believe our findings on the cooperation of two flagella bring more insights on zoospore swimming dynamics. It was also not from the literature about the role of each flagellum on the straight runs and on turning events, in particular. We show that although the energy is shared in comparable manner between both flagella, the anterior flagellum contributes more to zoospore speed. Zoospores actively change direction thanks to the sole beating of the anterior flagellum. We believe that anterior flagellum could be the main motor of zoospores that is in charge of generating speed, changing beating patterns from sinusoidal wave to power and recover stroke to quickly rotate the body, while the posterior flagellum might play a role in chemical/electrical sensing and providing an anchor-like turning point for zoospores, instead of acting like a rudder as previously hypothesized. These findings pave new ways for controlling the disease since now we can have different strategies on targeting one of the flagella.”

While I enjoyed reading the article, the authors should endeavour to make it clearer in certain places what has come before, and what is the novel component. For instance, the modelling seems to follow the arguments of https://doi.org/10.1063/1.3608240, and while this is cited elsewhere, I believe this should be stated very clearer in the model section. Secondly, there are a few sentences which mislead with respect to aspects of the novelty, for instance:1. "Despite the relevance of zoospore spreading in the epidemics of plant diseases, it is not known how these zoospores swim and steer with two opposite beating flagella." For the swimming this is clearly known, as the authors commented on in their previous work, it is the turning I believe that was not well characterised.2. "Here, we introduce a new type of microswimmer, named Phytophthora zoospores, which has two different flagella collaborating for unique swimming and turning mechanisms (Figure 1(a))." – you have presented work previously on the swimming of this organism.I believe the data on mastigoneme density is critical to one of your conclusions, and should be presented carefully, and I believe this conclusion about the anterior flagellum being the main driver for motility is probably not correct, more broadly – it certainly needs more careful treatment.

We thank the Reviewer for these recommendations. We have followed them and revised the manuscript accordingly. The Reviewer can find the answers to these recommendations in the previous questions.